# Efficient Image Restoration with State-Dependent Forward Diffusion

## Abstract

This paper proposes to perform image restoration through a state-dependent mean-reverting **fo**rward **d**iffusion (FoD) process. In contrast to traditional diffusion-based approaches that rely on a coupled forward-backward diffusion scheme, FoD directly learns image restoration through a single forward diffusion process, yielding a simple yet efficient framework. The core of FoD is a state-dependent stochastic differential equation (SDE) that involves a mean-reverting term in both the drift and diffusion functions. This mean-reverting structure drives the low-quality data toward the clean endpoint with controlled stochastic variation, therefore simulating a stochastic interpolation between source and target distributions. More importantly, FoD is analytically tractable and is trained using a simple stochastic flow matching objective, enabling few-step sampling during inference. The proposed FoD model, despite its simplicity, achieves strong overall performance on various image restoration tasks compared to representative diffusion, diffusion bridge, and flow matching approaches.

## 1 Introduction

Image restoration (IR) aims to recover high-quality (HQ) clean images from degraded low-quality (LQ) observations, such as rainy, hazy, low-light, or masked images. Unlike unconditional generation from pure noise, an LQ observation already contains rich structural information about the desired output (Luo et al., 2023a; Saharia et al., 2022b). This makes IR a conditional image-to-image problem, where an ideal stochastic restoration process should start from the LQ observation and progressively recover the HQ image.

Recent IR methods have increasingly adopted diffusion-based generative formulations to handle complex degradations and recover photo-realistic details (Saharia et al., 2022b; Kawar et al., 2022; Wang et al., 2024). However, many of these methods still inherit the forward-backward diffusion paradigm, where a forward process gradually perturbs clean data into noise, while a learned reverse process transforms noise back to data (Sohl-Dickstein et al., 2015; Ho et al., 2020; Song et al., 2021; Karras et al., 2022). Although effective, this iterative reverse-time denoising process often relies on a learned score function (Kawar et al., 2022; Saharia et al., 2022a) and can be computationally expensive for image restoration.

Diffusion bridge models provide a more direct transport formulation between source and target distributions, and have recently shown strong results in image-to-image translation and restoration tasks (Liu et al., 2023b; Zhou et al., 2024; Yue et al., 2024; Zhu et al., 2025). However, many of these methods rely on reverse-time formulations or additional bridge terms, which can complicate learning and sampling.

Another promising direction is flow matching (Lipman et al., 2022; Liu et al., 2022), which directly learns ordinary differential equations (ODEs) whose objective is to estimate a deterministic vector field from source distributions to the target distributions. While it has shown strong performance in generative modeling, this ODE formulation removes stochastic noise injection, which has been shown crucial in image restoration (Luo et al., 2023a;b). As a result, flow matching may struggle to capture high-frequency and perceptual details, leading to over-smoothed restoration results (Albergo et al., 2023a; Martin et al., 2024; Ohayon et al., 2024).

In this paper, we propose to perform image restoration through a single **fo**rward **d**iffusion (FoD) process. Our exploration starts from the mean-reverting stochastic differential equation (SDE) (Gillespie, 1996; Luo et al.,

2023a), where the data is stochastically driven toward a specified state characterized by a fixed mean and variance. Inspired by this, we construct a new variant of the mean-reverting SDE that adds mean-reversion to *both* the drift and diffusion functions as a state-dependent diffusion process. Here, we highlight the mean-reverting structure in the diffusion function, as it drives the process toward the noise-free mean state while reducing the stochastic perturbation near the target. By setting the mean to the target data, FoD naturally simulates the data transition between source and target distributions, without explicitly learning a reverse-time score model. In contrast to flow matching, FoD simulates SDEs rather than ODEs, preserving stochasticity during sampling and thereby providing a more suitable framework for image restoration.

We further demonstrate that FoD is analytically tractable and follows a multiplicative stochastic structure. Moreover, we show that the model can be learned by approximating the vector field from each intermediate noisy state to the final clean data, a process we refer to as *stochastic flow matching*. The result is a conceptually simple, yet effective training process. Based on the tractable solution and the stochastic flow matching objective, FoD enables few-step restoration with both Markov and non-Markov chains, providing a more efficient restoration procedure while maintaining competitive sample quality.

Our contributions are summarized below:

- We present FoD, a state-dependent mean-reverting forward SDE for image restoration. This SDE introduces the mean-reverting term to both the drift and diffusion functions, enabling stochastic restoration without explicitly learning a score-based reverse-time SDE process.

- We show that FoD is analytically tractable and follows a multiplicative stochastic structure, and that FoD can be learned through a simple stochastic flow matching objective.

- We develop efficient sampling strategies and empirically demonstrate that FoD enables few-step restoration while maintaining strong image restoration quality.

- We evaluate our model across various image restoration tasks and metrics, demonstrating strong overall performance compared with other representative generative restoration approaches.

## 2 Background

Given a source distribution $p_{\text{prior}}$ and an unknown target data distribution $p_{\text{data}}$, our goal is to build a probability path $\{p(x_t)\}_{t=0}^{T}$ that transports between the source distribution $p(x_0) = p_{\text{prior}}$ and the target distribution $p(x_T) = p_{\text{data}}$. In the image restoration setting, the source is often the distribution of degraded observations that already contain rich structural information about the target image.

**Diffusion Models**   Given a target data point $x_T \sim p_{\text{data}}$, diffusion models (Sohl-Dickstein et al., 2015; Ho et al., 2020) define a Markov chain forward process to progressively perturb the data into noise ($x_T \rightarrow x_0$) and then learn its reverse process to reconstruct the data ($x_0 \rightarrow x_T$). This coupled forward-backward process can be defined by stochastic differential equations (SDEs) (Song et al., 2021):

$$\mathrm{d}x_t = f(x_t, t)\,\mathrm{d}t + g(t)\,\mathrm{d}w_t, \tag{1}$$

$$\mathrm{d}x_t = \left[ f(x_t, t) - g(t)^2\,\nabla_x \log p_t(x_t) \right] \mathrm{d}t + g(t)\,\mathrm{d}\bar{w}_t, \tag{2}$$

for forward diffusion and its reverse-time processes, respectively. Here, $f(x, t)$ is the *drift* function and $g(t)$ is the *diffusion* function. $w$ and $\bar{w}$ are standard Wiener processes. We use $p_t(x_t)$ to denote the marginal probability density of $x_t$. The term $\nabla_x \log p_t(x_t)$, called the *score function*, is the sought-after objective in the backward (also called the reverse-time) SDE, which is often learned by a time-dependent neural network (Song et al., 2021) via score-matching. The training objective can also be converted to learn noise matching as in DDPMs (Ho et al., 2020). Moreover, the source distribution in diffusion models is often a Gaussian with a predefined mean and variance. Diffusion models typically require thousands of denoising steps to generate high-quality clean samples.

**Diffusion Bridge Models** enable stochastic transport between arbitrary distributions, by introducing Doob's $h$-transform (Doob, 1984; Särkkä & Solin, 2019) to guide the forward SDE to drift from data to a specified condition (Zhou et al., 2024; Yue et al., 2024). Recent works in diffusion bridge matching (Shi et al., 2023; Liu et al., 2023a) and bridge mixtures (Peluchetti, 2023) also construct optimal transport paths for data generation and have achieved impressive results. However, these approaches rely on either explicit bridge-consistency constraints or solving for a complex mixture of diffusion bridges, which complicates both the conceptual formulation and the practical implementation. In contrast, our method provides a conceptually simpler and analytically tractable SDE process for stochastic image restoration.

**Flow Matching** is a simple regression objective used for learning the velocity field $v(x_t, t)$ that transports a sample $x_t$ from the source distribution to the target distribution along the probability path $p(x_t)$ (Lipman et al., 2022; Liu et al., 2022). More specifically, flow matching models aim to learn the ordinary differential equation (ODE): $\mathrm{d}x_t = v(x_t, t)\,\mathrm{d}t$, where $x_0 \sim p_{\text{prior}}$ and the drift $v(x_t, t)$ transports samples from $x_0$ to $x_1 \sim p_{\text{data}}$. Here, each latent variable $x_t$ in the ODE path is drawn by linearly interpolating source and target data samples, i.e., $x_t = tx_1 + (1-t)x_0$. Then the training can be performed by uniformly sampling data pairs and timesteps and optimizing:

$$L_{\text{FM}}(\phi) = \mathbb{E}_{x_0, x_1, t \sim \mathcal{U}(0,1)} \left[ \|(x_1 - x_0) - v_\phi(x_t, t)\|^2 \right], \tag{3}$$

where $v_\phi(x_t, t)$ is a neural network approximating the true velocity field. Flow matching models provide a more direct learning procedure based on ODE paths. However, we observe that applying it to image-conditioned generation tasks, such as image restoration, leads to a significant performance drop due to its noise-free learning process (see Section 5.2). Moreover, it is worth noting that both diffusion models and flow matching models can be unified into the stochastic interpolants (Albergo et al., 2023a) framework.

## 3 Method

Our method defines a state-dependent mean-reverting forward SDE that evolves a degraded image toward its clean target, as illustrated in Figure 1. In the following, we first revisit the classical mean-reverting SDE (Section 3.1) and then introduce our state-dependent formulation and its closed-form solution in Section 3.2. Based on the solution, we propose a stochastic flow matching objective in Section 3.3, and present the efficient Markov and non-Markov chain sampling strategies in Section 3.4.

### 3.1 Preliminaries: Mean-reverting SDE

Our exploration starts from a mean-reverting SDE (Gillespie, 1996; Luo et al., 2023a) where the data is stochastically driven towards a state characterized by a specified mean $\mu$ and variance $\lambda^2$:

$$\mathrm{d}x_t = \theta_t \left( \mu - x_t \right) \mathrm{d}t + \sigma_t \, \mathrm{d}w_t, \tag{4}$$

where $\{\theta_t\}_{t=0}^T$ and $\{\sigma_t\}_{t=0}^T$ are positive mean-reversion and diffusion schedules, respectively. By coupling the schedules as $\sigma_t^2 / \theta_t = 2\,\lambda^2$ for all $t$, we obtain the following solution (Luo et al., 2023a)

$$x_t = \mu + \left( x_0 - \mu \right) e^{-\int_0^t \theta_z \, \mathrm{d}z} + \int_0^t \sigma_z \, e^{-\int_s^t \theta_s \, \mathrm{d}s} \, \mathrm{d}w_z. \tag{5}$$

As $t \to \infty$, the SDE converges to a stationary state $x_T \sim \mathcal{N}(x_T \mid \mu, \lambda^2)$. At first glance, this suggests a direct restoration process: one may set the initial state $x_0$ to the LQ image and the mean $\mu$ to the HQ target, so that the dynamics drive LQ observations toward HQ images. However, the terminal state $x_T$ remains noisy due to the non-vanishing stationary variance $\lambda^2$, and therefore does not converge exactly to the clean target.

Instead, Luo et al. (2023a) propose to use Equation 4 in the opposite direction, by setting $x_0$ and $\mu$ as the HQ and LQ images, respectively, so that the mean-reverting SDE naturally simulates image degradation. Then we could restore the HQ image by running backwards in time, i.e., the reverse-time SDE:

$$\mathrm{d}x_t = [\theta_t \left( \mu - x_t \right) - \sigma_t^2 \nabla_x \log p_t(x)] \, \mathrm{d}t + \sigma_t \, \mathrm{d}\bar{w}_t, \tag{6}$$

where the score function $\nabla_x \log p_t(x)$ can be computed using Equation 5. This coupled forward-backward process is referred to as the image restoration SDE (IR-SDE), illustrated in Figure 1(a).

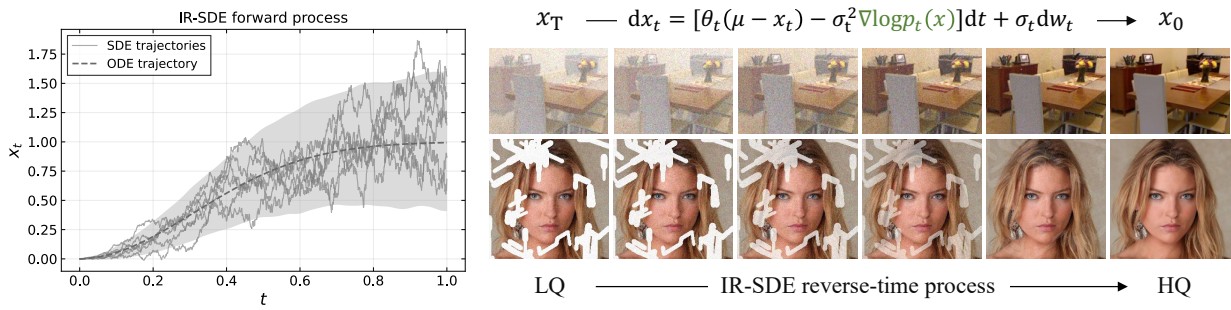

(a) IR-SDE forward process (left) and its reverse-time process for image restoration (right).

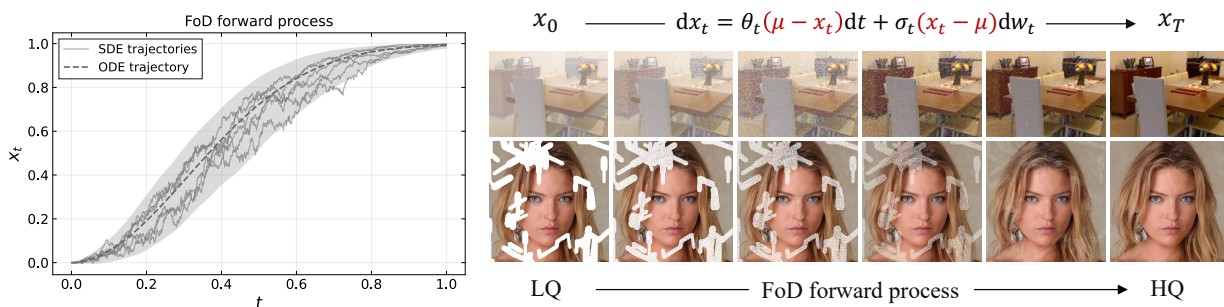

(b) The proposed FoD forward process for image restoration.

Figure 1: Comparison between IR-SDE and the proposed state-dependent mean-reverting forward diffusion (FoD) model. IR-SDE relies on a coupled forward-backward scheme, where restoration is performed by a reverse-time SDE with score estimation. In contrast, FoD directly recovers clean outputs through a single forward SDE. By introducing the mean-reversion term into *both* the drift and diffusion functions (marked in red), FoD preserves stochasticity and enables efficient high-quality image restoration.

## 3.2 State-Dependent Mean-Reverting SDE (FoD)

Note that the IR-SDE reverse-time process adds additional noise to the LQ image and requires a full simulation of the denoising process, making it computationally inefficient for image restoration. We therefore design a new SDE with mean-reversion terms in both the drift and diffusion functions, as

$$\mathrm{d}x_t = \theta_t \left( \mu - x_t \right) \mathrm{d}t + \sigma_t \left( x_t - \mu \right) \mathrm{d}w_t. \tag{7}$$

This is a state-dependent linear SDE with multiplicative noise, where the diffusion volatility increases in the beginning steps and then decreases to zero when $x_t$ converges to $\mu$, as illustrated in Figure 1(b). We typically use $x_t - \mu$ in the diffusion function such that this SDE simulates a reverse Wiener process as in diffusion models (Song et al., 2021). For image restoration, the noise $\mathrm{d}w_t$ is added independently at each pixel, meaning that this SDE is applied for images pixel-by-pixel, under the Itô interpretation.

We refer to this state-dependent mean-reverting process as FoD, and present its solution as follows:

**Proposition 3.1.** *Given an initial state $x_s$ at time $s < t$, the unique solution to the SDE Equation 7 is*

$$x_t = \left( x_s - \mu \right) \mathrm{e}^{- \int_s^t \left( \theta_z + \frac{1}{2} \sigma_z^2 \right) \mathrm{d}z + \int_s^t \sigma_z \mathrm{d}w_z} + \mu, \tag{8}$$

*where the stochastic integral is interpreted in the Itô sense and can be reparameterised as $\bar{\sigma}_{s:t} \, \epsilon$, where $\bar{\sigma}_{s:t} = \sqrt{\int_s^t \sigma_z^2 \, \mathrm{d}z}$, and $\epsilon \sim \mathcal{N}(0, I)$ is a standard Gaussian noise.*

The proof is provided in Appendix A. Positive $\theta$ and $\sigma$ schedules in Equation 8 introduce a strong exponential damping factor that drives $x_t$ toward $\mu$, thereby stabilizing the process over time. The analytical solution also

reveals that FoD is a special instance of stochastic interpolants (Albergo et al., 2023a), where the interpolation coefficient between $x_0$ and $\mu$ is stochastic and multiplicative. This connects FoD to the broader family of flow and diffusion formulations, while preserving its state-dependent stochastic path for image restoration (see Appendix C.2 for details).

In addition, the solution in Equation 8 shows that the stochastic flow field $\mu - x_t$ forms a Geometric Brownian motion (Ross, 2014) and yields the following corollary:

**Corollary 3.2.** *Under the same assumptions as in Proposition 3.1, the stochastic flow field $\mu - x_t$ satisfies the multiplicative stochastic structure. More precisely, it is log-normally distributed by*

$$\log |\mu - x_t| \sim \mathcal{N}\Big(\log |\mu - x_s| - \int_s^t \big(\theta_z + \frac{1}{2}\sigma_z^2\big)\,\mathrm{d}z, \int_s^t \sigma_z^2\,\mathrm{d}z\,I\Big). \tag{9}$$

This follows directly from Proposition 3.1 by rearranging $\mu - x_t$ to the left of Equation 8 and applying the logarithm to both sides, since the stochastic exponential factor is strictly positive and taking element-wise magnitudes therefore preserves the same multiplicative form. The subtractive form of the logarithm reflects that the flow field decays multiplicatively from its initial value with a stochastic exponential scaling.

*Notational Clarifications:* Since $\mu - x_t$ can be either positive or negative element-wise, we write the logarithmic terms in Equation 9 using the magnitude $|\mu - x_t|$. The sign information is preserved: for a given sample, the element-wise sign of $\mu - x_t$ remains consistent across all times $t$, as the FoD transition only multiplies $\mu - x_s$ by a strictly positive stochastic exponential factor (see Equation 8). In addition, we let $\bar{m}_{s:t} = -\int_s^t (\theta_z + \frac{1}{2}\sigma_z^2)\,\mathrm{d}z$, $\bar{m}_t = \bar{m}_{0:t}$, and $\bar{\sigma}_t = \bar{\sigma}_{0:t}$ in the rest of the paper to simplify the notation.

## 3.3 Learning FoD by Stochastic Flow Matching

We now describe how to learn the proposed FoD process. Specifically, given a paired training sample $(x_0, \mu)$, where $x_0$ is the degraded LQ image and $\mu$ is the clean HQ target, the closed-form solution in Proposition 3.1 allows us to sample any intermediate state $x_t$ directly:

$$x_t = (x_0 - \mu)e^{\bar{m}_t + \bar{\sigma}_t \epsilon} + \mu, \qquad \epsilon \sim \mathcal{N}(0, I). \tag{10}$$

This gives a stochastic path from the degraded LQ observation $x_0$ to HQ image $\mu$. Since FoD directly evolves samples toward the target, the natural learning target at each intermediate state is the residual direction $\mu - x_t$, which points from the current state to the clean endpoint. This target is analogous to the velocity field in flow matching, but is learned over stochastic FoD trajectories rather than deterministic ODE paths. Therefore, we can optimize the following stochastic flow matching objective:

$$\mathcal{L}_{\text{SFM}}(\phi) = \mathbb{E}_{x_0, \mu, t, \epsilon}\left[\|(\mu - x_t) - f_\phi(x_t, t)\|_2^2\right], \tag{11}$$

where $f_\phi(x_t, t)$ is a neural network. The detailed FoD training process is illustrated in Algorithm 1. Intuitively, this objective guides the model how to move each noisy intermediate state toward the clean target while being exposed to the stochastic perturbations induced by the SDE. In Appendix B, we provide a variational motivation for this objective, showing that the FoD transition induces a log-residual likelihood, for which the proposed regression loss serves as a first-order surrogate rather than an exact variational objective. In addition, we also show that the first-order approximation can be loose during early training (see Table A1).

## 3.4 Efficient Sampling for Image Restoration

After training, the clean target can be estimated from any intermediate state by

$$\hat{\mu}_\phi(x_t, t) = x_t + f_\phi(x_t, t). \tag{12}$$

A straightforward way to generate samples is to solve the SDE in Equation 7 with the Euler–Maruyama method, as shown in Algorithm 2. While this provides a standard discretization of the FoD process, it typically requires a large number of discretization steps and thus is inefficient for image restoration.

---

**Algorithm 1** FoD Training

**Require:** $p_{\text{prior}}, p_{\text{data}}$, model $f_\phi$
1: **repeat**
2:     $x_0 \sim p_{\text{prior}}, \mu \sim p_{\text{data}}$
3:     $\epsilon \sim \mathcal{N}(0, I), t \sim \text{Uniform}(\{1, \ldots, T\})$
4:     $x_t = (x_0 - \mu) \, e^{\bar{m}_t + \bar{\sigma}_t \epsilon} + \mu$
5:     Take gradient descent step on
6:         $\nabla_\phi \|(\mu - x_t) - f_\phi(x_t, t)\|^2$
7: **until** converged

**Algorithm 2** FoD Sampling

**Require:** $p_{\text{prior}}$, time interval $\Delta t$, model $f_\phi$
1: $x_0 \sim p_{\text{prior}}$
2: **for** $t = 0, \ldots, T - 1$ **do**
3:     $\epsilon \sim \mathcal{N}(0, I)$
4:     $\Delta x = \theta_t f_\phi(x_t, t) \cdot \Delta t - \sigma_t f_\phi(x_t, t) \cdot \sqrt{\Delta t} \, \epsilon$
5:     $x_{t+1} = x_t + \Delta x$
6: **end for**
7: **return** $x_T$

---

**Algorithm 3** Markov Chain Sampling

**Require:** $p_{\text{prior}}$, step size $k$, model $f_\phi$
1: $x_0 \sim p_{\text{prior}}$
2: **for** $t = 0, k, 2k, \ldots, T$ **do**
3:     $\epsilon \sim \mathcal{N}(0, I)$
4:     $\hat{\mu} = x_t + f_\phi(x_t, t)$
5:     $x_{t+k} = \left( \boxed{x_t} - \hat{\mu} \right) e^{\bar{m}_{t:t+k} \, + \, \epsilon \cdot \bar{\sigma}_{t:t+k}} + \hat{\mu}$
6: **end for**
7: **return** $x_T$

**Algorithm 4** Non-Markov Chain Sampling

**Require:** $p_{\text{prior}}$, step size $k$, model $f_\phi$
1: $x_0 \sim p_{\text{prior}}$
2: **for** $t = 0, k, 2k, \ldots, T$ **do**
3:     $\epsilon \sim \mathcal{N}(0, I)$
4:     $\hat{\mu} = x_t + f_\phi(x_t, t)$
5:     $x_{t+k} = \left( \boxed{x_0} - \hat{\mu} \right) e^{\bar{m}_{t+k} + \epsilon \cdot \bar{\sigma}_{t+k}} + \hat{\mu}$
6: **end for**
7: **return** $x_T$

---

Fortunately, the analytical FoD solution in Proposition 3.1 naturally enables few-step sampling. Specifically, let us define a coarse time grid $t \in \{0, k, 2k, \ldots, T\}$, where $k$ is the step size. At each timestep, we first predict the clean target $\hat{\mu}_\phi$ using Equation 12, and then directly sample the next state $x_{t+k}$ from the closed-form transition. This leads to a $k$ times faster inference than the standard Euler–Maruyama method.

Moreover, we could consider two variants: Markov chain sampling and non-Markov chain sampling. The Markov chain sampling uses the current state $x_t$ as the starting point of each transition. In contrast, the non-Markov chain sampling anchors each transition at the initial degraded observation $x_0$:

$$x_{t+k} = (x_0 - \hat{\mu}_\phi) e^{\bar{m}_{t+k} + \epsilon \cdot \bar{\sigma}_{t+k}} + \hat{\mu}_\phi. \tag{13}$$

The Markov sampling recursively refines the current state, while the non-Markov sampling reduces error accumulation by repeatedly referring to the initial observation. These two strategies are summarized in Algorithms 3 and 4, and their empirical behavior is analyzed in Section 5.1.

# 4 Experiments

We evaluate our method across different image restoration tasks and metrics, and compare with representative iterative generative restoration methods, including diffusion, diffusion-bridge, and flow matching approaches[1].

## 4.1 Implementation and Setup

We use a U-Net (Ronneberger et al., 2015; Dhariwal & Nichol, 2021) architecture for flow prediction in all tasks. Attention layers are removed for efficient training and testing, similar to IR-SDE (Luo et al., 2023a;b). We choose the commonly used cosine and linear schedules (Nichol & Dhariwal, 2021) for $\theta_t$ and $\sigma_t$, respectively, and normalize $\sigma_t^2$ to sum to 1 to ensure numerical stability under multiplicative noise perturbation. The number of sampling steps is fixed to 100 for all tasks. We use the AdamW (Loshchilov & Hutter, 2017) optimizer with parameters $\beta_1 = 0.9$ and $\beta_2 = 0.99$. The training requires $500\,000$ iterations with a learning rate of $10^{-4}$. All models are trained on an A100 GPU with 40 GB of memory for approximately 1.5 days.

---

[1]We provide additional results and implementation details in Appendix D.

Table 1: Quantitative comparison of FoD with other representative diffusion, diffusion bridge, and flow matching restoration approaches on deraining and inpainting tasks.

| Method | Category | Image deraining | | | | Face inpainting | | | |
|---|---|---|---|---|---|---|---|---|---|
| | | PSNR↑ | SSIM↑ | LPIPS↓ | FID↓ | PSNR↑ | SSIM↑ | LPIPS↓ | FID↓ |
| U-Net | *Baseline* | 29.12 | 0.882 | 0.153 | 57.55 | 27.97 | 0.889 | 0.097 | 58.78 |
| IR-SDE | *Diffusion* | 31.65 | 0.904 | 0.047 | 18.64 | 29.83 | 0.904 | 0.045 | 26.30 |
| GOUB | *Diffusion bridge* | 31.96 | 0.903 | 0.046 | 18.14 | 29.81 | 0.916 | 0.039 | 23.39 |
| UniDB | | 32.05 | 0.904 | 0.045 | 17.65 | 30.01 | 0.917 | 0.038 | 23.16 |
| ReFlow | *Flow matching* | 28.36 | 0.871 | 0.152 | 64.81 | 29.84 | 0.912 | 0.065 | 38.65 |
| PMRF | | 29.01 | 0.857 | 0.173 | 69.25 | **30.45** | 0.901 | 0.082 | 55.40 |
| FoD | *Ours* | **32.56** | **0.925** | **0.038** | **14.10** | 30.28 | **0.923** | **0.029** | **16.12** |

Table 2: Quantitative comparison of FoD with other representative diffusion, diffusion bridge, and flow matching restoration approaches on low-light enhancement and dehazing tasks.

| Method | Category | Low-light enhancement | | | | Image dehazing | | | |
|---|---|---|---|---|---|---|---|---|---|
| | | PSNR↑ | SSIM↑ | LPIPS↓ | FID↓ | PSNR↑ | SSIM↑ | LPIPS↓ | FID↓ |
| U-Net | *Baseline* | 20.51 | 0.808 | 0.162 | 75.84 | 22.88 | 0.906 | 0.065 | 15.65 |
| IR-SDE | *Diffusion* | 20.45 | 0.787 | 0.129 | 47.28 | 25.25 | 0.906 | 0.060 | 8.33 |
| GOUB | *Diffusion bridge* | 19.29 | 0.775 | 0.148 | 50.44 | 25.31 | 0.908 | 0.048 | 8.21 |
| UniDB | | 20.18 | 0.796 | 0.128 | 45.61 | 25.65 | 0.896 | 0.051 | 8.29 |
| ReFlow | *Flow matching* | 19.62 | 0.767 | 0.221 | 91.93 | 20.84 | 0.864 | 0.081 | 23.53 |
| PMRF | | 19.32 | 0.753 | 0.189 | 81.59 | 22.45 | 0.868 | 0.092 | 24.09 |
| FoD | *Ours* | **21.61** | **0.819** | **0.105** | **41.31** | **26.57** | **0.932** | **0.033** | **8.14** |

**Evaluation Metrics.** In all experiments, the Learned Perceptual Image Patch Similarity (LPIPS) (Zhang et al., 2018) and Fréchet Inception Distance (FID) (Heusel et al., 2017) are reported to evaluate the perceptual fidelity and overall visual quality. Additionally, Peak Signal-to-Noise Ratio (PSNR) and SSIM (Wang et al., 2004) are also included to evaluate pixel-level and structural similarity.

## 4.2 Comparison with Other Approaches

We evaluate our method on four IR tasks: 1) image deraining on the Rain100H dataset (Yang et al., 2017), 2) dehazing on the RESIDE-6k dataset (Qin et al., 2020), 3) low-light enhancement on LOL (Wei et al., 2018), and 4) face inpainting on CelebA-HQ (Karras et al., 2017).

**Comparison Approaches.** Our empirical comparison focuses on iterative generative restoration methods, since our goal is to study how a state-dependent forward SDE can improve stochastic iterative restoration and few-step sampling. We select IR-SDE (Luo et al., 2023a) as the main comparison method to evaluate the performance gap between forward-backward and forward-only schemes for diffusion-based restoration. We further compare against two diffusion bridge models, GOUB (Yue et al., 2024) and UniDB (Zhu et al., 2025), both of which leverage mean-reverting bridges for image-conditioned generation. Moreover, we implement two flow matching based approaches, Rectified flow (Liu et al., 2022; Liu, 2022) and posterior-mean rectified flow (PMRF) (Ohayon et al., 2024), that learn ODEs and also enable the model to generate images with a single forward process. For a fair comparison, all these methods use 100 sampling steps to gradually restore the HQ images. In addition, a U-Net model, using the same architecture as our FoD, is trained with the $\ell_2$ loss as a CNN baseline on all tasks for reference.

**Results.** The quantitative comparisons on four IR tasks are reported in Table 1 and Table 2. The proposed FoD achieves the best or competitive results on most reported metrics across the evaluated datasets in comparison to other diffusion, diffusion bridge, and flow-based approaches. Compared to the U-Net baseline, IR-SDE successfully improves the results on perceptual metrics (LPIPS and FID), proving the effectiveness of the forward-backward based diffusion IR schemes. By leveraging bridges, GOUB and UniDB further improve the IR-SDE results across most tasks and metrics, but are still consistently outperformed by our FoD. We also

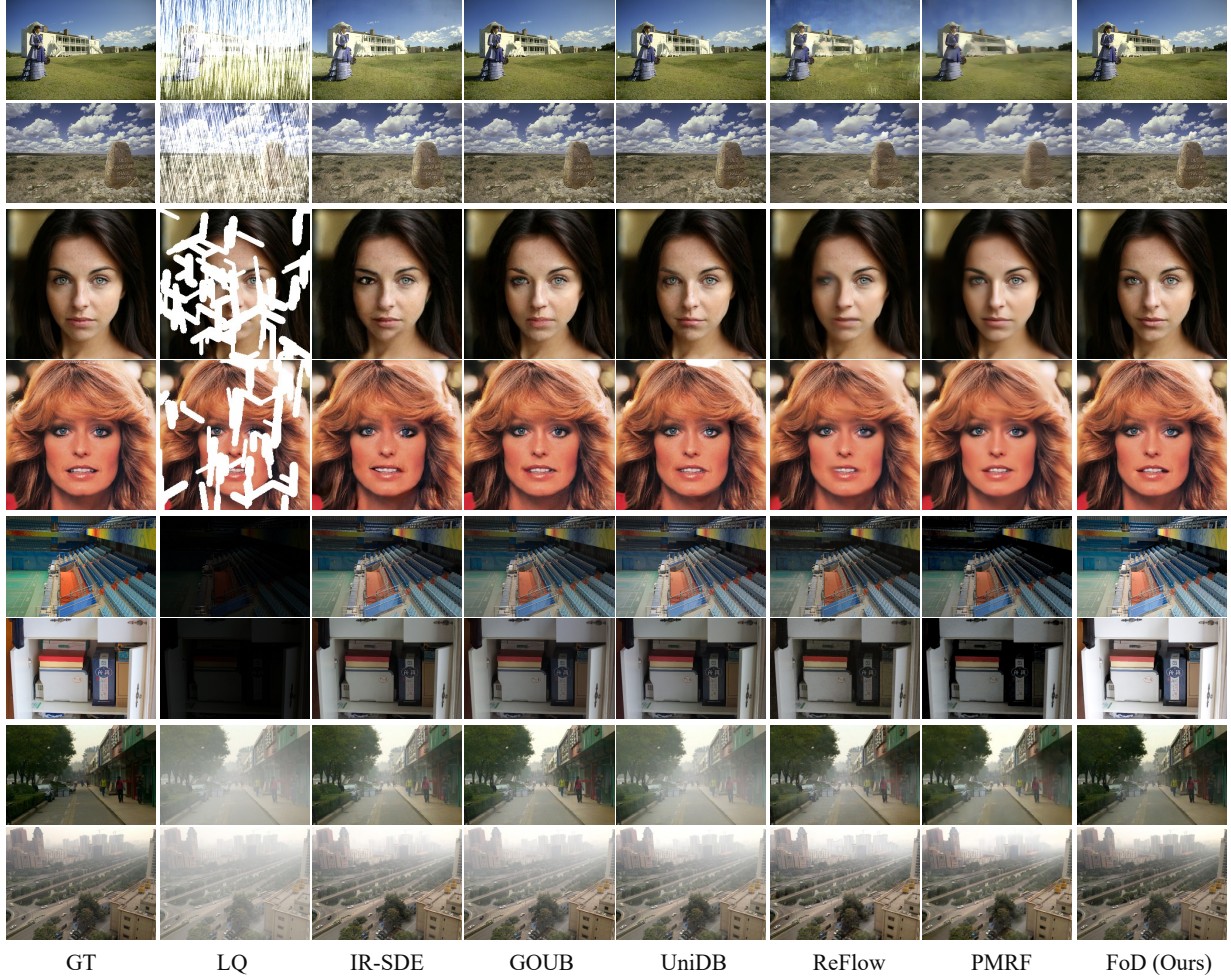

| GT | LQ | IR-SDE | GOUB | UniDB | ReFlow | PMRF | FoD (Ours) |

Figure 2: Comparison of FoD with other representative diffusion, diffusion-bridge, and flow-matching baselines on four IR tasks including image deraining, inpainting, low-light enhancement, and dehazing.

observe that flow-based approaches, such as ReFlow and PMRF, generally underperform FoD on perceptual metrics and most restoration settings, although PMRF obtains a higher PSNR on face inpainting. While PMRF improves the PSNR results for flow matching-based IR, the performance gain potentially comes from the two-stage training strategy and the small noise injection in the initial state of rectified flow. We note that these improvements are not uniform across all metrics. For example, on face inpainting, PMRF obtains a higher PSNR than FoD, while FoD remains competitive on the other metrics. Thus, our results should be interpreted as strong overall performance rather than uniform superiority on every metric.

We also provide visual comparisons in Figure 2, showing that our FoD produces the most realistic and high-fidelity results. In particular, while deterministic approaches without noise injection (ReFlow and PMRF) tend to generate overly smooth outputs (see e.g. the left eye area in the face inpainting case), the proposed FoD model consistently produces sharper and more detailed images.

## 5 Discussion and Analysis

In this section, we analyze the key properties of FoD, including its few-step sampling, the role of stochastic noise injection, the behavior of the learned diffusion process, and the effect of different noise schedules.

Image Deraining

Low-light enhancement

Image Dehazing

Image Inpainting

--- Baseline    ● FoD w/ MC    ■ FoD w/ NMC

Figure 3: Comparison of different sampling approaches with pretrained FoD models on four image restoration tasks. The baseline is the Euler–Maruyama method with 100 sampling steps.

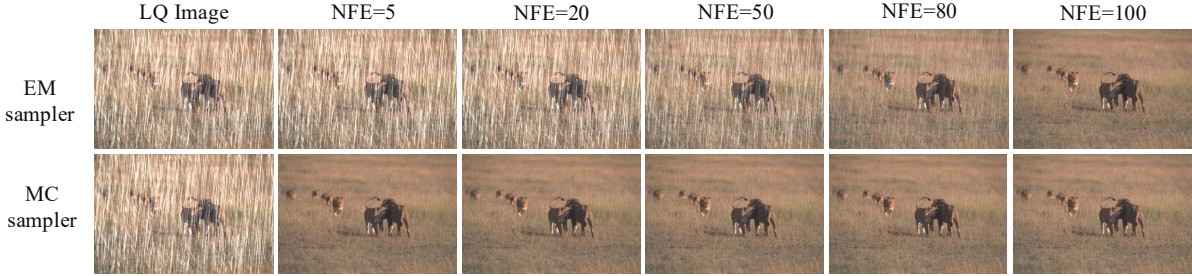

Figure 4: Qualitative comparison between the standard diffusion (Euler–Maruyama) sampler and our efficient Markov chain sampler. All images are generated by sampling from a pretrained FoD model on image deraining.

## 5.1 Fast Sampling

As discussed in Section 3.4, FoD naturally supports efficient sampling by directly using its analytical transition in Equation 8. Table 3 compares the standard 100-step Euler–Maruyama (EM) sampler with our 10-step

Table 3: Comparison of different sampling methods. Here, 'FoD w/ EM' denotes the *100-step* Euler–Maruyama sampling method, while 'FoD w/ MC' and 'FoD w/ NMC' denote *10-step* Markov and non-Markov chain sampling, respectively.

| Task | Metric | FoD w/ EM | FoD w/ MC | FoD w/ NMC |
|---|---|---|---|---|
| Deraining | PSNR↑ | 32.56 | 33.27 | **33.63** |
| | SSIM↑ | 0.925 | 0.934 | **0.941** |
| | LPIPS↓ | **0.038** | 0.039 | 0.041 |
| | FID↓ | **14.10** | 15.14 | 15.64 |
| Low-light | PSNR↑ | 21.61 | **23.12** | 23.05 |
| | SSIM↑ | 0.819 | 0.850 | **0.855** |
| | LPIPS↓ | 0.105 | **0.093** | 0.098 |
| | FID↓ | 41.31 | **32.37** | 47.87 |
| Dehazing | PSNR↑ | 26.57 | 26.76 | **26.77** |
| | SSIM↑ | 0.932 | 0.938 | **0.941** |
| | LPIPS↓ | 0.033 | **0.031** | 0.032 |
| | FID↓ | **8.14** | 10.07 | 10.31 |
| Inpainting | PSNR↑ | 30.28 | 31.02 | **31.32** |
| | SSIM↑ | 0.923 | 0.933 | **0.938** |
| | LPIPS↓ | **0.029** | 0.031 | 0.038 |
| | FID↓ | **16.12** | 18.06 | 23.28 |

Table 4: Ablation experiment to illustrate the effectiveness of noise injection. The noise-free variant is obtained by setting $\sigma_t = 0$ for all timesteps.

| Task | Metric | FoD w/o noise | FoD |
|---|---|---|---|
| Deraining | PSNR↑ | 29.44 | **32.56** |
| | SSIM↑ | 0.880 | **0.925** |
| | LPIPS↓ | 0.132 | **0.038** |
| | FID↓ | 59.54 | **14.10** |
| Low-light | PSNR↑ | 19.96 | **21.61** |
| | SSIM↑ | 0.772 | **0.819** |
| | LPIPS↓ | 0.193 | **0.105** |
| | FID↓ | 86.21 | **41.31** |
| Dehazing | PSNR↑ | 24.18 | **26.57** |
| | SSIM↑ | 0.876 | **0.932** |
| | LPIPS↓ | 0.048 | **0.033** |
| | FID↓ | 13.80 | **8.14** |
| Inpainting | PSNR↑ | 29.94 | **30.28** |
| | SSIM↑ | 0.922 | **0.923** |
| | LPIPS↓ | 0.065 | **0.029** |
| | FID↓ | 38.78 | **16.12** |

Markov-chain (MC) and non-Markov-chain (NMC) samplers. Despite using only one tenth of the sampling steps, both efficient samplers achieve competitive or even better distortion metrics. Intuitively, the MC and NMC samplers use the closed-form FoD transition to move between time points and re-estimate the clean endpoint $\hat{\mu}$ at each step, which makes the sampling procedure closer to the training objective and thereby alleviates the accumulation of discretization errors along the sampling trajectory. In particular, the NMC sampler obtains the best PSNR and SSIM on most tasks, showing its advantage in preserving image structure by anchoring each transition at the initial degraded observation. We further analyze the effect of the number of sampling steps in Figure 3, where the 100-step EM sampler is used as the baseline. Both MC and NMC samplers perform well in the low-step regime, especially with 5–20 sampling steps. The NMC sampler generally achieves stronger PSNR and SSIM at very small NFE, while its LPIPS and FID may degrade when using more steps, suggesting that repeatedly referring to the initial observation helps preserve structure but may limit perceptual refinement. In contrast, the MC sampler updates the current state recursively and shows more stable perceptual behavior across different NFEs. However, MC/NMC can also slightly degrade perceptual metrics such as LPIPS and FID compared with the 100-step EM sampler in some cases.

Figure 4 provides a qualitative comparison between the standard EM sampler and our proposed MC sampler. The EM sampler requires many small denoising steps to gradually remove degradations, whereas the MC sampler produces visually plausible restoration results with substantially fewer steps. Overall, these results show that FoD enables effective few-step image restoration, with 10 steps serving as a practical trade-off between efficiency, structural fidelity, and perceptual quality.

## 5.2 Effectiveness of Noise Injection

In this section, we explore the importance of noise injection in image restoration. To this end, we train a noise-free variant of FoD by setting $\sigma_t = 0$ for all $t$, which reduces FoD to a deterministic flow matching model as described in Section C.2. For a fair comparison, we keep the $\theta$ schedule unchanged for this noise-free variant. Quantitative results in Table 4 show that removing noise injection causes a consistent performance drop across all tasks and metrics. The degradation is especially clear on perceptual metrics: for example, on deraining, LPIPS increases from 0.038 to 0.132 and FID increases from 14.10 to 59.54. Similar trends are observed on low-light enhancement and inpainting, indicating that stochasticity is important for recovering high-frequency and perceptual details. The qualitative comparisons in Figure 5 also show that stochastic noise injection helps recover sharper and more realistic details. These results demonstrate the effectiveness and importance of noise injection in image restoration.

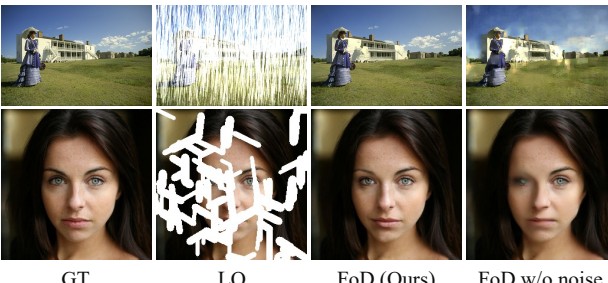

GT      LQ      FoD (Ours)      FoD w/o noise

Figure 5: Comparison of FoD with its noise-free variant on image deraining and face inpainting tasks. The full FoD model restores sharper and more realistic details, such as clearer background structures in deraining and more faithful facial textures in inpainting. By contrast, removing the stochastic diffusion term leads to over-smoothed predictions and visible artifacts in heavily degraded regions.

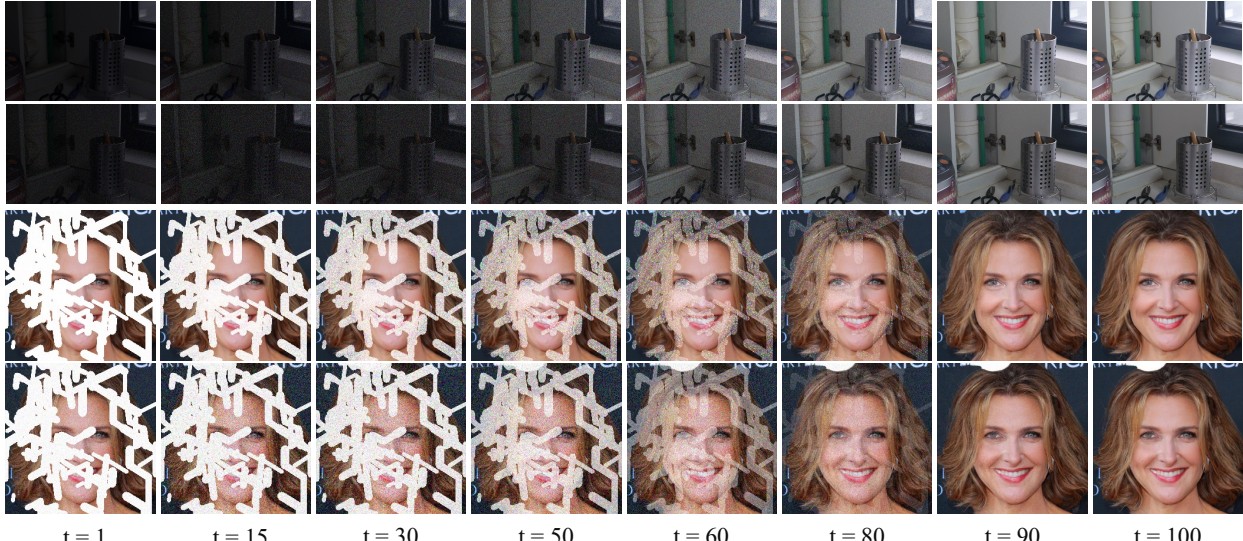

t = 1    t = 15    t = 30    t = 50    t = 60    t = 80    t = 90    t = 100

Figure 6: Comparison of the sampling dynamics of FoD (top row) and GOUB (Yue et al., 2024) (bottom row) across different image restoration tasks. Both methods introduce stochastic perturbations and progressively recover clean images, while FoD tends to concentrate the perturbations more on degraded regions.

## 5.3 Illustration of the Diffusion Process

To better understand the sampling behavior of FoD, we visualize intermediate states generated by the trained model at different timesteps in Figure 6. For each restoration task, the top row shows the FoD process and the bottom row shows the GOUB Yue et al. (2024) diffusion bridge process. Both methods gradually inject noise into intermediate states and then recover clean images as the process approaches the terminal time. However, their noise patterns are noticeably different. GOUB tends to perturb the entire image, while FoD mainly introduces stochasticity in the degraded regions, such as low-light areas, haze-corrupted regions, and masked facial regions. This behavior is consistent with the analytical FoD solution, where the stochastic perturbation is modulated by the residual term $\mu - x_0$. Regions with larger differences between the LQ input and the HQ target naturally receive stronger perturbations, whereas already reliable regions are less affected. As a result, FoD focuses more on restoring degraded content rather than reconstructing the whole image from scratch, making the process well suited for image restoration.

## 5.4 Choice of Noise Schedules

We use the commonly adopted cosine schedule for $\theta_t$ and linear schedule for $\sigma_t$ in our main experiments. To analyze the effect of schedule choices, we further compare different combinations of $\theta_t$ and $\sigma_t$ on the deraining task. The training curves in Figure 7 show that FoD remains robust under different choices of $\theta_t$ and $\sigma_t$ schedules. The quantitative results in Table 5 further show that all variants outperform the IR-SDE baseline, demonstrating the robustness of the proposed FoD framework.

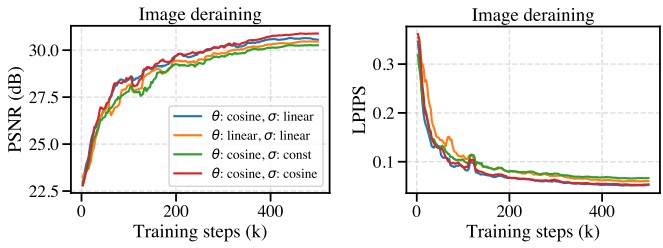

Figure 7: Training curves of different $\theta$ and $\sigma$ schedules.

Table 5: Analysis of different choices of $\theta$ and $\sigma$ schedules on image deraining.

| Method | PSNR↑ | SSIM↑ | LPIPS↓ | FID ↓ |
|---|---|---|---|---|
| IR-SDE (baseline) | 31.65 | 0.904 | 0.047 | 18.64 |
| linear $\theta$, linear $\sigma$ | 32.48 | 0.918 | 0.045 | 16.11 |
| cosine $\theta$, const $\sigma$ | 32.22 | 0.906 | 0.046 | 17.34 |
| cosine $\theta$, linear $\sigma$ | 32.56 | 0.925 | 0.038 | 14.10 |
| cosine $\theta$, cosine $\sigma$ | 32.73 | 0.931 | 0.039 | 15.12 |

Among different choices, using a constant $\sigma_t$ leads to slightly worse results, especially in perceptual metrics. Applying cosine schedules to both $\theta_t$ and $\sigma_t$ achieves the best PSNR and SSIM, while the cosine-$\theta_t$ and linear-$\sigma_t$ setting obtains the best LPIPS and FID. We therefore adopt cosine $\theta_t$ and linear $\sigma_t$ as the default schedule, as it provides a favorable trade-off between distortion and perceptual quality.

## 6 Related Work

Recent image restoration (IR) methods have increasingly adopted generative formulations, including diffusion models, diffusion bridges, and flow matching, to recover photo-realistic images from degraded observations (Saharia et al., 2022b; Kawar et al., 2022; Liu et al., 2023b; Albergo et al., 2023a). Diffusion-based restoration methods typically formulate restoration as a conditional denoising problem and have achieved strong performance across inpainting, super-resolution, deraining, and real-world restoration tasks (Lugmayr et al., 2022; Yue et al., 2023; Wang et al., 2024; Liu et al., 2024; Shi et al., 2024; Luo et al., 2024). Their stochastic sampling process provides useful generative priors and improves perceptual quality, but most methods still rely on a forward-backward diffusion scheme or an iterative reverse-time denoising process, which can make sampling computationally expensive.

Diffusion bridge and Schrödinger bridge based methods provide a more direct stochastic transport formulation between degraded and clean image distributions (Liu et al., 2023b; Li et al., 2023; Su et al., 2022; Zhou et al., 2024; Yue et al., 2024). By constructing paths between source and target distributions, these methods better align with the conditional nature of image restoration. However, they often require bridge-specific mechanisms, such as bridge consistency constraints, Doob's $h$-transforms, or stochastic-control objectives (Zhu et al., 2025), which can complicate both the formulation and implementation.

Flow matching and stochastic interpolants offer another direct transport perspective by learning vector fields or constructing probability paths between source and target distributions (Lipman et al., 2022; Liu et al., 2022; Albergo et al., 2023a). Recent restoration methods based on these ideas enable efficient generation (Albergo et al., 2023b; Ohayon et al., 2024; Ben-Hamu et al., 2024; Martin et al., 2024). However, deterministic flow-based formulations remove stochasticity, which may limit their ability to recover high-frequency and perceptual details, leading to over-smoothed restorations in image restoration settings. While stochastic interpolants provide a general framework for combining deterministic transport and stochastic perturbations, they do not by themselves prescribe an IR-specific stochastic path.

In contrast, FoD defines a state-dependent mean-reverting SDE tailored to image restoration. It can be interpreted as a particular multiplicative stochastic interpolant, where the interpolation coefficient is induced by the FoD transition and the stochastic perturbation is modulated by the residual to the mean state. By introducing this mean-reverting residual into both the drift and diffusion terms, FoD preserves stochastic noise injection for perceptual detail recovery while avoiding an explicit reverse-time score process, yielding a simple, analytically tractable, and efficient stochastic restoration framework.

## 7 Conclusion

This paper presents FoD, a state-dependent mean-reverting forward SDE framework for efficient image restoration. FoD introduces mean reversion into both the drift and diffusion functions, enabling a single

stochastic process that directly restores degraded observations without learning a reverse-time SDE. We show that FoD is analytically tractable, can be trained with a simple stochastic flow matching objective, and naturally supports few-step Markov and non-Markov sampling. Experiments on multiple image restoration tasks demonstrate that FoD achieves strong performance compared to representative diffusion, diffusion-bridge, and flow matching baselines, highlighting its effectiveness and efficiency for high-quality image restoration.

**Limitations and future work.** Although FoD achieves strong results on several image restoration tasks, this work mainly focuses on paired and relatively well-defined degradations. Its behavior under real-world degradations, unpaired settings, and broader image-conditioned generation tasks remains to be further investigated. Moreover, FoD uses independent Gaussian perturbations for analytical tractability. While this leads to a closed-form transition and efficient sampling, it may not fully capture complex degradation distributions. In addition, our main experiments only use a lightweight UNet backbone for efficiency. In experiments, the evaluation does not quantify the variability induced by stochastic sampling. This limitation is especially relevant for FID on small test sets such as LOL, where the estimator can have higher variance. Strong regression-based restoration networks are not included in the current comparison. In future work, we will further explore adaptive schedule designs, larger architectures, latent space generation, and extensions to more challenging restoration and generation settings.

## Broader Impact Statement

The proposed framework is designed to improve the efficiency and quality of image restoration, which may benefit applications such as low-light photography, degraded image enhancement, and other visual systems requiring efficient recovery from corrupted observations. Its few-step sampling strategy may also reduce the computational cost of diffusion-based restoration.

However, as a generative restoration model, FoD may hallucinate plausible but incorrect details in heavily degraded or masked regions, especially in tasks such as face inpainting. Therefore, restored images should not be treated as ground truth in high-stakes settings such as forensics, surveillance, identity verification, legal evidence, or medical imaging. In addition, high-quality restoration models may also be misused to manipulate visual content or remove imperceptible protective signals, such as watermarks, adversarial perturbations, or unlearnable examples. This work focuses on standard paired restoration benchmarks and does not study these security-related scenarios. We therefore encourage future work to evaluate the robustness, misuse risks, and deployment constraints of stochastic restoration models in sensitive applications.

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

## Appendix

## A Proof to Proposition 3.1 and Corollary 3.2

**Proposition 3.1.** *Given an initial state $x_s$ at time $s < t$, the unique solution to the SDE equation 7 is*

$$x_t = (x_s - \mu)\, \mathrm{e}^{-\int_s^t \left(\theta_z + \frac{1}{2}\sigma_z^2\right)\mathrm{d}z + \int_s^t \sigma_z \mathrm{d}w_z} + \mu, \tag{A1}$$

*where the stochastic integral is interpreted in the Itô sense and can be reparameterised as $\bar{\sigma}_{s:t}\,\epsilon$, where $\bar{\sigma}_{s:t} = \sqrt{\int_s^t \sigma_z^2\, \mathrm{d}z}$, and $\epsilon \sim \mathcal{N}(0, I)$ is a standard Gaussian noise.*

**Corollary 3.2.** *Under the same assumptions as in Proposition 3.1, the stochastic flow field $\mu - x_t$ satisfies the multiplicative stochastic structure. More precisely, it is log-normally distributed by*

$$\log|\mu - x_t| \sim \mathcal{N}\Big(\log|\mu - x_s| - \int_s^t \left(\theta_z + \frac{1}{2}\sigma_z^2\right)\mathrm{d}z, \int_s^t \sigma_z^2\, \mathrm{d}z\, I\Big). \tag{A2}$$

*Proof.* Recall the FoD process from Equation 7:

$$\mathrm{d}x_t = \theta_t\,(\mu - x_t)\,\mathrm{d}t + \sigma_t\,(x_t - \mu)\,\mathrm{d}w_t. \tag{A3}$$

To solve this SDE, we introduce a new variable $y_t = x_t - \mu$ and replace it into Equation A3, which typically yields a Geometric Brownian motion (Ross, 2014) on $y_t$, given by

$$\mathrm{d}y_t = -\theta_t y_t\,\mathrm{d}t + \sigma_t y_t\,\mathrm{d}w_t. \tag{A4}$$

To simplify the notation, we use $y$ rather than $y_t$ in all the following equations. This equation can be solved by applying Itô's formula:

$$\begin{aligned}
\mathrm{d}\psi(y, t) = {} & \frac{\partial \psi}{\partial t}(y, t)\,\mathrm{d}t + \frac{\partial \psi}{\partial y}(y, t) f(y, t)\,\mathrm{d}t \\
& + \frac{1}{2}\frac{\partial^2 \psi}{\partial y^2}(y, t) g(t)^2\,\mathrm{d}t \\
& + \frac{\partial \psi}{\partial y}(y, t) g(t)\,\mathrm{d}w,
\end{aligned} \tag{A5}$$

where $\psi(y, t) = \ln|y|$ is a surrogate differentiable function. By substituting $f(y, t)$ and $g(t)$ with the drift and the diffusion functions in (A4), we obtain

$$\mathrm{d}\psi(y, t) = -(\theta_t + \frac{\sigma_t^2}{2})\,\mathrm{d}t + \sigma_t\,\mathrm{d}w. \tag{A6}$$

Then we can solve $y_t$ conditioned on $y_s$, by integrating both sides:

$$\ln|y_t| - \ln|y_s| = -\int_s^t (\theta_z + \frac{\sigma_z^2}{2})\,\mathrm{d}z + \int_s^t \sigma_z \mathrm{d}w(z) \tag{A7}$$

where the stochastic integral follows a Gaussian distribution, i.e., $\int_s^t \sigma_z\, \mathrm{d}w(z) \sim \mathcal{N}\big(0, \int_s^t \sigma_z^2 \mathrm{d}z\big)$, then we can rewrite:

$$\ln|y_t| = \ln|y_s| - (\bar{\theta}_{s:t} + \frac{\bar{\sigma}_{s:t}^2}{2}) + \bar{\sigma}_{s:t}\epsilon_{s \to t}, \quad \epsilon_{s \to t} \sim \mathcal{N}(0, I), \tag{A8}$$

where $\bar{\theta}_{s:t} = \int_s^t \theta_z\, \mathrm{d}z$, $\bar{\sigma}_{s:t}^2 = \int_s^t \sigma_z^2\, \mathrm{d}z$, and $\bar{\sigma}_{s:t} = \sqrt{\bar{\sigma}_{s:t}^2}$. By replacing $y_t$ with the original $x_t - \mu$, we have the following

$$\ln|x_t - \mu| = \ln|x_s - \mu| - (\bar{\theta}_{s:t} + \frac{\bar{\sigma}_{s:t}^2}{2}) + \bar{\sigma}_{s:t}\epsilon_{s \to t}, \tag{A9}$$

which leads to the following log-normal distribution:

$$\log|\mu - x_t| \sim \mathcal{N}\Big(\log|\mu - x_s| - \int_s^t \big(\theta_z + \frac{1}{2}\sigma_z^2\big)\,\mathrm{d}z, \int_s^t \sigma_z^2\,\mathrm{d}z\,I\Big), \tag{A10}$$

which gives the Corollary 3.2. Note that the sign information is preserved: for a given sample, the element-wise sign of $\mu - x_t$ remains consistent across all times $t$. Then, applying the exponential function to both sides yields

$$(x_t - \mu) = (x_s - \mu)\mathrm{e}^{-(\bar{\theta}_{s:t} + \frac{\bar{\sigma}_{s:t}^2}{2}) + \bar{\sigma}_{s:t}\epsilon_{s\to t}} \tag{A11}$$

which is the solution to the SDE, and thus we complete the proof.

Note that we consider the FoD process under the Itô interpretation. The schedules $\theta_t$ and $\sigma_t$ are assumed to be deterministic, positive, and integrable over the time interval of interest. The driving process $w_t$ is a standard Wiener process, and the noise is applied independently across pixels. Under these assumptions, FoD is a state-dependent linear SDE with a unique strong solution. We also note that these assumptions provide a tractable stochastic restoration path, but may not fully capture complex real-world degradations. $\qquad\square$

## B   A Variational Explanation of the Stochastic Flow Matching Loss

Here, we show that our stochastic flow matching loss can also be viewed as a stable surrogate of the log-residual likelihood implied by the multiplicative FoD transition, while avoiding numerical issues caused by logarithms of small or signed residuals. Specifically, we define the FoD model in discrete time, as $p_\phi(x_{0:T})$, a joint distribution with learnable transitions starting at $x_0$, as

$$p_\phi(x_{0:T}) = p_{\text{prior}}(x_0) \prod_{t=0}^{T-1} p_\phi(x_{t+1} \mid x_t), \quad x_0 \sim p_{\text{prior}}. \tag{A12}$$

By setting the transition kernel $p_\phi(x_{t+1}|x_t)$ to be in the same log-Gaussian form as Equation 8, the training can be performed by minimizing the negative log-likelihood of $p_\phi(x_T)$, which leads to the following objective:

$$\mathbb{E}_p\Big[\sum_{t=0}^{T-1} D_{KL}\big(p(x_{t+1} \mid x_t, x_T) \,\|\, p_\phi(x_{t+1} \mid x_t)\big)\Big]. \tag{A13}$$

The proof is provided in B.1. During training, we use $\mu$ as the clean endpoint condition. The finite-time process approaches this endpoint approximately, with the residual $\mu - x_t$ controlled by the schedules, rather than converging exactly to $\mu$. Then, the conditional distribution $p(x_{t+1}|x_t, x_T = \mu)$ is tractable and can be computed using Proposition 3.1. By letting the functions $f_\mu(x_t) = \mu - x_t$ and $f_\phi(x_t, t) = \hat{\mu}_\phi - x_t$ denote the ground truth and the model prediction of the stochastic flow field, respectively, we transform the distributions in Equation A13 from SDE states to stochastic flow fields. Note that this transformation, i.e., from $p(\cdot|x_t, \mu)$ to $p(\cdot|f_\mu(x_t))$, holds because its Jacobian determinant equals one. Instead of Equation A13, we can therefore minimize the KL divergence between two stochastic flow (log-residual) distributions:

$$\mathbb{E}_p\Big[\sum_{t=0}^{T-1} D_{KL}(p(f_\mu(x_{t+1}) \mid f_\mu(x_t)) \,\|\, p(f_\phi(x_{t+1}, t) \mid f_\phi(x_t, t)))\Big]. \tag{A14}$$

where we define $p_{f_\mu}(x_{t+1} \mid x_t) \coloneqq p(f_\mu(x_{t+1}) \mid f_\mu(x_t))$ and $p_{f_\phi}(x_{t+1} \mid x_t) \coloneqq p(f_\phi(x_{t+1}, t) \mid f_\phi(x_t, t))$ as the log-residual distributions, i.e., $\log(\mu - x_t)$. Combining this with Corollary 3.2, we obtain the following surrogate objective:

$$\begin{aligned} L_{\text{SFM}}(\phi) &\coloneqq \mathbb{E}_{\mu, x_t}\Big[\,\|\log|\mu - x_t| - \log f_\phi(x_t, t)\|^2\,\Big] \\ &\approx \mathbb{E}_{\mu, x_t}\Big[\|(\mu - x_t) - f_\phi(x_t, t)\|^2\Big], \end{aligned} \tag{A15}$$

where the approximation follows from a first-order Taylor expansion close to the optimum. However, we also note that the first-order approximation can be loose during early training. Please refer to B.2 for more details. This objective is linear and thus leads to a simple and numerically stable training process.

## B.1 Proof of the KL Divergence

The KL divergence of Equation A13 can be derived as follows:

$$\tilde{L} := -\log p_\phi(x_T) \tag{A16}$$

$$= -\log \int p_\phi(x_{0:T}) \, dx_{0:T-1} \tag{A17}$$

$$= -\log \int \frac{p_\phi(x_{0:T}) q(x_{0:T-1} \mid x_T)}{q(x_{0:T-1} \mid x_T)} \, dx_{0:T-1} \tag{A18}$$

$$= -\log \mathbb{E}_{q(x_{0:T-1}|x_T)} \left[ \frac{p_\phi(x_{0:T})}{q(x_{0:T-1} \mid x_T)} \right] \tag{A19}$$

$$\leq \underbrace{\mathbb{E}_{q(x_{0:T-1}|x_T)} \left[ -\log \frac{p_\phi(x_{0:T})}{q(x_{0:T-1} \mid x_T)} \right]}_{\text{negative evidence lower bound (ELBO)}} \qquad \text{(Jensen's Inequality)} \tag{A20}$$

$$= \mathbb{E}_{q(x_{0:T-1}|x_T)} \left[ -\log \frac{p(x_0) \prod_{t=1}^{T} p_\phi(x_t \mid x_{t-1})}{\prod_{t=1}^{T} q(x_{t-1} \mid x_t)} \right] \tag{A21}$$

$$= \mathbb{E}_{q(x_{0:T-1}|x_T)} \left[ -\log p(x_0) - \sum_{t=1}^{T} \log \frac{p_\phi(x_t \mid x_{t-1})}{q(x_{t-1} \mid x_t)} \right] \tag{A22}$$

$$= \mathbb{E}_{q(x_{0:T-1}|x_T)} \left[ -\log p(x_0) - \sum_{t=1}^{T-1} \log \frac{p_\phi(x_t \mid x_{t-1})}{\underbrace{q(x_t \mid x_{t-1}, x_T)} \cdot \frac{q(x_t \mid x_T)}{q(x_{t-1} \mid x_T)}}_{\text{Bayes' rule on } q(x_{t-1}|x_t, x_T)} - \log \frac{p_\phi(x_T \mid x_{T-1})}{q(x_{T-1} \mid x_T)} \right] \tag{A23}$$

$$= \mathbb{E}_{q(x_{0:T-1}|x_T)} \left[ -\log p(x_0) - \sum_{t=1}^{T-1} \log \frac{p_\phi(x_t \mid x_{t-1})}{q(x_t \mid x_{t-1}, x_T)} - \log \frac{q(x_{T-1} \mid x_T)}{q(x_0 \mid x_T)} - \log \frac{p_\phi(x_T \mid x_{T-1})}{q(x_{T-1} \mid x_T)} \right] \tag{A24}$$

$$= \mathbb{E}_{q(x_{0:T-1}|x_T)} \left[ -\log \frac{p(x_0)}{q(x_0 \mid x_T)} - \sum_{t=1}^{T-1} \log \frac{p_\phi(x_t \mid x_{t-1})}{q(x_t \mid x_{t-1}, x_T)} - \log p_\phi(x_T \mid x_{T-1}) \right] \tag{A25}$$

$$= D_{KL}(q(x_0 \mid x_T) \,||\, p(x_0)) \qquad (D_{KL}(p\|q) = \mathbb{E}[-\log \frac{p}{q}]) \tag{A26}$$

$$+ \sum_{t=1}^{T-1} D_{KL}(q(x_t \mid x_{t-1}, x_T) \,||\, p_\phi(x_t \mid x_{t-1})) - \mathbb{E}_{q(x_{T-1}|x_T)} \left[ \log p_\phi(x_T \mid x_{T-1}) \right], \tag{A27}$$

where the first term can be ignored since it doesn't have trainable parameters, and the third term can be merged to the final stochastic flow matching objective.

## B.2 Derivation of the Log-Residual Distribution Based Loss

**Remark.** For two log-normal distributions with $\log p_1 \sim \mathcal{N}(\mu_1, \sigma_1^2)$ and $\log p_2 \sim \mathcal{N}(\mu_2, \sigma_2^2)$, the KL divergence between them is given by

$$D_{\text{KL}}(p_1 \| p_2) = \frac{(\mu_1 - \mu_2)^2}{2\sigma_2^2} + \frac{\sigma_1^2}{2\sigma_2^2} + \ln \frac{\sigma_2}{\sigma_1} - \frac{1}{2}. \tag{A28}$$

This result is for scalars but can be naturally extended to high-dimensional cases. In the following, we will use it to derive our stochastic flow matching objective.

More specifically, given the KL divergence

$$\mathbb{E}_p \left[ \sum_{t=0}^{T-1} D_{KL}(p(f_\mu(x_{t+1}) \mid f_\mu(x_t) \,\|\, p(f_\phi(x_{t+1}, t) \mid f_\phi(x_t, t)) \right] \tag{A29}$$

and the truth that $f_\phi(x_t, t)$ approximates $\mu - x_t$. Since the two transitions are log-normally distributed and share the same parameters $\theta_t$ and $\sigma_t$ as in Equation 9, we can initially obtain a log space loss based on the Remark equation A28, as

$$L := \mathbb{E}_{\mu \sim p_{\text{data}}, x_t \sim q(x_t|x_0, \mu)} \left[ \frac{1}{2\sigma_{t+1}^2} \| \log f_\mu(x_t, t) - \log f_\phi(x_t, t) \|^2 \right]. \tag{A30}$$

However, this would run into issues when $\mu - x_t \leq 0$. Though this can be alleviated using absolute values and adding a small additive term $\epsilon$, it complicates the learning and is still unstable.

To address it, we assume that, after some training, $f_\phi(x_t, t)/f_\mu(x_t, t) = 1 + \delta$ where $|\delta| \ll 1$. The first-order Taylor approximation is then

$$\log f_\phi(x_t, t) - \log f_\mu(x_t, t) = \log(1 + \delta) \approx \frac{f_\phi(x_t, t) - f_\mu(x_t, t)}{f_\mu(x_t, t)}. \tag{A31}$$

Since the denominator does not depend on the parameters, we obtain our simplified loss function by omitting all non-trainable weights:

$$L := \mathbb{E}_{\mu \sim p_{\text{data}}, x_t \sim q(x_t|x_0, \mu)} \left[ \| f_\mu(x_t, t) - f_\phi(x_t, t) \|^2 \right], \tag{A32}$$

which is the proposed stochastic flow matching objective.

Note that the first-order Taylor approximation is not valid during the early stages of training, when the predicted flow is typically far from the ground truth. In such cases, Equation A15 no longer corresponds to an exact KL objective, but instead serves as a surrogate loss for directly learning the flow. Nevertheless, we emphasize that this surrogate objective can empirically achieve strong performance and simplify the optimization. This is conceptually analogous to denoising objectives and those used in score matching (not exact KL objectives, but have still proven effective in practice). Table A1 below tracks both the approximation error $(\log(1 + \delta) - \delta)$ and magnitude of the higher-order terms $(-\frac{1}{2}\delta^2)$. As one can see, the approximation is loose during the early stages of training but becomes valid $(\approx 0.1)$ after $\sim 50{,}000$ iterations.

Table A1: Tracking the approximation error and magnitude of the higher-order terms.

| Training steps | Approximation error | Magnitude |
|---|---|---|
| 1,000 | 0.404 | 0.333 |
| 10,000 | 0.183 | 0.224 |
| 50,000 | 0.105 | 0.108 |
| 100,000 | 0.095 | 0.086 |

## C   Additional Discussion

### C.1   Maximum Likelihood Estimation

Following DDPMs (Ho et al., 2020), both our training and sampling are implemented with discrete times, which can of course be converted into continuous times, as in Score SDEs (Song et al., 2021), but that requires $\theta$ and $\sigma$ schedules to be integrable. Below, we further show that the solution to FoD also allows us to compute the maximum likelihood:

Given the clean data $\mu$, assuming that there exists an optimal forward transition from $z_t$ to $z_{t+1}$, where $z_t = |\mu - x_t|$. In other words, we want to maximize the likelihood of $p(z_{t+1} \mid z_t)$, which is a log-normal distribution as illustrated in Corollary 3.2, and its density is given by

$$p(z_{t+1} \mid z_t) = \frac{1}{z_{t+1}\sigma_{t+1}\sqrt{2\pi}} \exp\left( -\frac{1}{2\sigma_{t+1}^2} \left[ \ln z_{t+1} - \ln z_t + \left(\theta_{t+1} + \frac{\sigma_{t+1}^2}{2}\right) \right]^2 \right). \tag{A33}$$

Then, we can minimize the negative log-likelihood:

$$-\ln p(z_{t+1} \mid z_t) = \ln z_{t+1} + \frac{1}{2\sigma_{t+1}^2}\left[\ln z_{t+1} - lnz_t + \left(\theta_{t+1} + \frac{\sigma_{t+1}^2}{2}\right)\right]^2 + \ln(\sigma_{t+1}\sqrt{2\pi}), \qquad \text{(A34)}$$

which can be solved by setting the gradient to 0, as

$$\nabla_{z_{t+1}} - \ln p(z_{t+1} \mid z_t) = \frac{1}{z_{t+1}}\left[1 + \frac{1}{\sigma_{t+1}^2}\left(\ln z_{t+1} - lnz_t + \left(\theta_{t+1} + \frac{\sigma_{t+1}^2}{2}\right)\right)\right] = 0. \qquad \text{(A35)}$$

Since $z_{t+1}$ is not zero, the optimal solution $z_{t+1}^*$ is obtained according to

$$z_{t+1} = z_t e^{-(\theta_{t+1} + \frac{\sigma_{t+1}^2}{2}) - \sigma_{t+1}^2}. \qquad \text{(A36)}$$

Recall that $\mu - x_t$ has the same sign for all times $t$. Replacing $z_t$ with $|\mu - x_t|$ gives the following:

$$(\mu - x_{t+1})^* = (\mu - x_t)e^{-(\theta_{t+1} + \frac{\sigma_{t+1}^2}{2}) - \sigma_{t+1}^2}, \qquad \text{(A37)}$$

which is the optimal forward flow from $x_{t+1}$ to $\mu$.

Based on it, we can get the maximum likelihood learning objective:

$$L = \left\|(\mu - x_{t+1})^* - \mathbb{E}[\mu_\phi - x_{t+1}]\right\|^2, \qquad \text{(A38)}$$

$$L = \left\|x_{t+1}^* - \mathbb{E}_\phi[x_{t+1} \mid x_t]\right\|^2, \qquad \text{(A39)}$$

where $\mathbb{E}_\phi[x_{t+1} \mid x_t]$ is the expectation given $x_t$ in discrete time: $\mathbb{E}_\phi[x_{t+1} \mid x_t] = x_t + \mathrm{d}x_t$.

$$\begin{aligned}
\mathbb{E}\big[\mu_\phi - x_{t+1}\big] &= (\mu_\phi - x_t)e^{-(\theta_{t+1} + \frac{\sigma_{t+1}^2}{2}) + \frac{\sigma_{t+1}^2}{2}} \\
&= (\mu_\phi - x_t)e^{-\theta_{t+1}}.
\end{aligned} \qquad \text{(A40)}$$

Combining equation A37 and equation A40 predicts $x_{t+1}$ in the optimal path. This maximum likelihood-based loss function performs similarly to stochastic flow matching and can be potentially used in future work.

## C.2   Connection to Prior Work

In this section, we establish the theoretical connections between FoD and two closely related prior works: stochastic interpolants (SI) (Albergo et al., 2023a) and flow matching (FM) (Lipman et al., 2022). SI provides a unified stochastic framework to bridge two arbitrary distributions, while FM formulates a deterministic transport map between two distributions via an ODE.

### C.2.1   Stochastic Interpolants

Let us recall the solution in Equation 8 of the FoD process. By setting the initial state to $x_0$ and rearranging the equation, we obtain a stochastic process in the interpolant form:

$$I(t, x_0, \mu) = x_0\,\alpha_t + \mu\,(1 - \alpha_t), \qquad \text{(A41)}$$

$$\alpha_t = e^{-\int_0^t \left(\theta_z + \frac{1}{2}\sigma_z^2\right)\mathrm{d}z + \int_0^t \sigma_z \mathrm{d}w_z}. \qquad \text{(A42)}$$

Here, $I(t, x_0, \mu) = x_t$ satisfies the boundary conditions of a stochastic interpolant, with randomness introduced via $\mathrm{d}w$. FoD can thus be viewed as a powerful instantiation of SI, distinguished by two properties: multiplicative log-normal interpolation and a state-dependent stochastic path from $x_0$ to $\mu$. This formulation allows noise to be gradually added and subsequently removed within a single forward process. This perspective helps unify FoD with a broader class of generative frameworks.

### C.2.2 Flow Matching

We consider a deterministic version of the FoD process in Equation 7, i.e., omitting the diffusion term or setting $\sigma_t = 0$ for all times. This gives a mean-reverting ODE that bridges two distributions without noise injection, as $dx_t = \theta_t (\mu - x_t) \, dt$ with solution $x_t = (x_s - \mu) \, e^{-\int_s^t \theta_z \, dz} + \mu$. Setting $s$ to 0 and rewriting this solution yields an interpolation between $x_0$ and $\mu$:

$$x_t = x_0 \, \alpha_t + \mu \, (1 - \alpha_t), \quad \alpha_t = e^{-\int_0^t \theta_z \, dz}, \tag{A43}$$

which forms a similar transportation path as in flow matching but with a special velocity field given by $\theta_t (\mu - x_t)$. We can then learn the drift, resulting in a conditional flow matching objective:

$$L_{\text{CFM}} \coloneqq \mathbb{E}_{\mu, x_t} \left[ \| (\mu - x_t) - f_\phi(x_t, t) \|^2 \right] \tag{A44}$$

$$= \mathbb{E}_{\mu, x_t} \left[ \| \alpha_t(\mu - x_0) - f_\phi(x_t, t) \|^2 \right], \tag{A45}$$

which is a deterministic form of the stochastic flow matching in Equation 11. In practice, we can learn the target displacement $\mu - x_0$ directly and define the $\alpha$ schedule to be linear, i.e., decreasing from 1 to 0, in which case this mean-reverting ODE becomes flow matching with a straight-line path (Lipman et al., 2022; Liu et al., 2022) exactly. In other words, our primary FoD model can also be regarded as a stochastic extension of flow matching models.

### C.3 Expected Terminal Residual

Here, we further clarify the finite-time convergence behavior of FoD. Let $y_t = x_t - \mu$ denote the residual to the clean target. From the closed-form FoD solution, the terminal residual is

$$y_T = y_0 \exp \left( -\int_0^T (\theta_t + \frac{1}{2}\sigma_t^2) \, dt + \int_0^T \sigma_t \, dw_t \right). \tag{A46}$$

Since the stochastic integral satisfies $\int_0^T \sigma_t \, dw_t \sim \mathcal{N}\left(0, \int_0^T \sigma_t^2 \, dt\right)$, we obtain

$$\mathbb{E}[x_T - \mu] = (x_0 - \mu) \exp \left( -\int_0^T \theta_t \, dt \right). \tag{A47}$$

Thus, the expected terminal residual is bounded by

$$\| \mathbb{E}[x_T - \mu] \| \leq \| x_0 - \mu \| \exp(-\int_0^T \theta_t \, dt). \tag{A48}$$

This shows that the mean residual decays exponentially with the cumulative mean-reversion strength.

Moreover, the terminal stochasticity is controlled by the diffusion schedule. From the log-residual space used in Corollary 3.2, we can derive the log-space residual variance

$$\text{Var}(\log |x_T - \mu|) = \int_0^T \sigma_t^2 \, dt. \tag{A49}$$

In our implementation, this quantity is normalized to 1 for numerical stability. Therefore, the finite-time terminal state should be interpreted as an approximate convergence to the clean target with controlled stochastic variation, rather than exact convergence to $\mu$.

## D  Additional Experimental Details

### D.1  Implementation and Datasets

During training, we set the batch size to 16 and randomly crop paired images into $256 \times 256$ patches for all restoration tasks. Following common practice in image restoration, we keep the original resolution of each test image during evaluation, without resizing or cropping. Since FoD is stochastic, we use a fixed evaluation protocol: for each LQ test image, we generate one corresponding HQ image and compute all metrics on this generated output. We do not perform best-of-$N$ selection or average over multiple random samples. For perceptual metrics, FID is computed using the standard PyTorch-FID implementation[2], and LPIPS is computed using the Image Quality Assessment (IQA) toolbox[3].

We set the number of diffusion steps to 100 for all tasks. Practically, we choose to use a discrete time implementation for our method, where we let $\bar{\theta}_t = \int_0^t \theta_z \, \mathrm{d}z \approx \sum \theta_t \Delta t$ and $\bar{\sigma}_t^2 = \int_s^t \sigma_z^2 \, \mathrm{d}z \approx \sum \sigma_t^2 \Delta t$. To ensure that FoD converges to the clean data $\mu$ with controlled variation, we let the deterministic exponential term at the terminal state be a smaller value, i.e. $\mathrm{e}^{-\int_0^t (\theta_s + \frac{1}{2}\sigma_s^2) \, \mathrm{d}s} = \delta = 0.001$. Solving it leads to an updated time interval $\Delta t = \frac{\log \delta}{\int_0^t (\theta_s + \frac{1}{2}\sigma_s^2) \, \mathrm{d}s}$. Note that the small value of the deterministic exponential term makes the expected terminal residual close to zero, while the stochastic exponential factor introduces a small nonzero terminal variance. This also indicates that large values are required for the FoD's drift and diffusion coefficients, i.e. $\theta_t$ and $\sigma_t$, ensuring the stochasticity and that sufficient noise is added to the sampling process.

In addition, the details of image restoration datasets are listed below:

- *Deraining*: collected from the Rain100H (Yang et al., 2017) dataset containing 1800 images for training and 100 images for testing.

- *Dehazing*: collected from the RESIDE-6k (Qin et al., 2020) dataset which has mixed indoor and outdoor images with 6000 images for training and 1000 images for testing.

- *Low-light enhancement*: collected from the LOL (Wei et al., 2018) dataset containing 485 images for training and 15 images for testing.

- *Face inpainting*: we use CelebaHQ as the training dataset and divide 100 images with 100 thin masks from RePaint (Lugmayr et al., 2022) for testing.

### D.2 Efficiency Comparison

To substantiate the efficiency claim, we provide an additional comparison on the deraining task in terms of wall-clock inference time, number of function evaluations (NFE), parameter count, and PSNR. The wall-clock time is measured as the average inference time per image under the same evaluation setting. Since different groups of methods use different implementations and backbones, we also report the parameter count to clarify the source of computational cost. Specifically, IR-SDE, GOUB, and UniDB use the same UNet architecture with 137.15M parameters, while ReFlow, PMRF, and FoD are implemented under another shared guided-diffusion UNet framework with 73.45M parameters. PMRF additionally uses a posterior-mean predictor with 36.22M parameters.

As shown in Table A2, FoD with the 100-step sampler has similar wall-clock cost to ReFlow under the same guided-diffusion backbone, while achieving substantially higher PSNR. More importantly, the proposed few-step MC sampler reduces the NFE from 100 to 10 and the inference time from 3.85s to 0.43s, while further improving PSNR from 32.56 to 33.27. These results indicate that the efficiency gain of FoD mainly comes from its few-step sampling enabled by the closed-form transition, rather than from an architecture-independent wall-clock advantage over all baselines.

### D.3 Effect of Sampling Steps for ODE Baselines

To examine whether the 100-step evaluation protocol disproportionately penalizes ODE-based baselines, we further evaluate ReFlow (Liu et al., 2022) on image deraining with different numbers of sampling steps. ReFlow is used as a representative flow-matching ODE baseline. As shown in Table A3, reducing the number

---

[2]https://github.com/mseitzer/pytorch-fid
[3]https://github.com/chaofengc/iqa-pytorch

Table A2: Efficiency comparison in terms of wall-clock inference time, NFE, parameter count, and PSNR on the deraining task. IR-SDE, GOUB, and UniDB use the same UNet architecture, while ReFlow, PMRF, and FoD use another shared guided-diffusion (Dhariwal & Nichol, 2021) UNet framework.

| Method | Wall-clock time | NFE | Parameter count | PSNR↑ |
|---|---|---|---|---|
| IR-SDE | 11.59s | 100 | 137.15M | 31.65 |
| GOUB | 20.11s | 100 | 137.15M | 31.96 |
| UniDB | 20.17s | 100 | 137.15M | 32.05 |
| ReFlow | 3.86s | 100 | 73.45M | 28.36 |
| PMRF | 3.89s | 100 | 36.22M+73.45M | 29.01 |
| FoD (Ours) | 3.85s | 100 | 73.45M | 32.56 |
| FoD w/ MC (Ours) | 0.43s | 10 | 73.45M | 33.27 |

of sampling steps from 100 to 10 does not improve the restoration quality. Instead, the results remain nearly unchanged, with a slight degradation in PSNR and FID as the number of steps decreases. This suggests that the 100-step setting does not unfairly penalize the ODE baseline in terms of restoration quality. In contrast, the proposed FoD fast samplers in Figure 3 benefit from the closed-form stochastic transition and can achieve strong restoration quality with substantially fewer steps. Therefore, the advantage of FoD is not simply due to choosing a favorable number of sampling steps, but comes from the proposed state-dependent stochastic formulation and its efficient few-step sampling strategy.

Table A3: Comparison of the ODE baseline ReFlow and FoD on image deraining using different numbers of sampling steps.

| Method | PSNR↑ | SSIM↑ | LPIPS↓ | FID↓ |
|---|---|---|---|---|
| ReFlow ($T = 100$) | 28.36 | 0.871 | 0.152 | 64.81 |
| ReFlow ($T = 50$) | 28.34 | 0.871 | 0.153 | 64.88 |
| ReFlow ($T = 20$) | 28.32 | 0.870 | 0.153 | 64.99 |
| ReFlow ($T = 10$) | 28.31 | 0.869 | 0.153 | 65.06 |
| FoD ($T = 100$) | 32.56 | 0.925 | 0.038 | **14.10** |
| FoD ($T = 10$) | **33.27** | **0.934** | 0.039 | 15.14 |

### D.4 Ablation with Attention-Based Backbones

To examine whether FoD can work with attention-based architectures, we conduct a small-scale backbone ablation by comparing the default UNet, a UNet with attention layers, and a ViT-based backbone under the same FoD framework. As shown in Table A4, adding attention layers to the UNet slightly improves PSNR, SSIM, and LPIPS compared with the default FoD-UNet, while producing a slightly worse FID. This suggests that FoD is compatible with attention modules and can benefit from additional non-local modeling capacity. The ViT-based backbone, however, performs worse than the UNet-based variants in this setting. This indicates that directly replacing the UNet with a transformer backbone may require more careful architectural design, hyperparameter tuning, or larger-scale training. We therefore view scalable attention-heavy architectures for FoD as a promising direction for future work.

### D.5 Additional Results on JPEG Compression

To further examine the behavior of FoD beyond spatially localized degradations, we add an additional experiment on JPEG compression artifacts. Specifically, we use DIV2K (Agustsson & Timofte, 2017) as the training data and set the JPEG quality factor to 10. The test set is LIVE1 (Sheikh, 2005). JPEG compression introduces structured artifacts that are not purely local in the pixel domain, and therefore serves as a useful stress test for the residual-dependent stochastic perturbation used in FoD. As shown in Table A5,

Table A4: Small-scale ablation of FoD with different backbones.

| Method | PSNR↑ | SSIM↑ | LPIPS↓ | FID↓ |
|---|---|---|---|---|
| FoD-ViT | 29.69 | 0.912 | 0.041 | 19.71 |
| FoD-UNet | 30.28 | 0.923 | 0.029 | **16.12** |
| FoD-UNet w/ attn | **30.44** | **0.925** | **0.027** | 16.40 |

FoD remains effective under JPEG compression. The proposed FoD variants achieve the best PSNR, SSIM, LPIPS, or FID among the compared methods. In particular, FoD w/ MC obtains the best PSNR and LPIPS, FoD w/ NMC obtains the best SSIM, and the 100-step FoD sampler obtains the best FID. These results suggest that FoD is not limited to localized spatial corruptions and can also handle JPEG compression artifacts, although a broader evaluation on more frequency-domain degradations remains useful future work.

Table A5: Results on JPEG compression artifact removal.

| Method | PSNR↑ | SSIM↑ | LPIPS↓ | FID↓ |
|---|---|---|---|---|
| U-Net | 28.25 | 0.7926 | 0.300 | 76.77 |
| IR-SDE | 28.35 | 0.7947 | 0.299 | 76.75 |
| GOUB | 28.45 | 0.7954 | 0.295 | 76.47 |
| UniDB | 28.47 | 0.7958 | 0.297 | 76.46 |
| ReFlow | 28.46 | 0.7991 | 0.270 | 71.27 |
| PMRF | 28.47 | 0.8010 | 0.282 | 71.29 |
| FoD | 28.60 | 0.8002 | 0.271 | **70.98** |
| FoD w/ MC | **28.64** | 0.8014 | **0.268** | 71.14 |
| FoD w/ NMC | 28.61 | **0.8026** | 0.269 | 71.16 |

## D.6 Failure Cases

Although FoD achieves strong overall performance across several restoration tasks, it can still fail in challenging cases. Figure A1 shows two representative examples. In face inpainting, FoD generally reconstructs plausible facial structures, but it may introduce local semantic artifacts when the masked region covers highly structured components such as eyes. Since these regions require both accurate geometry and semantic consistency, the model can occasionally generate unnatural details or slight asymmetry. In image dehazing, FoD can effectively recover most scene content, but may leave residual haze or produce inaccurate local contrast in regions with strong illumination changes or complex background structures. These cases suggest that FoD can still be challenged by highly semantic missing regions and spatially non-uniform degradations. These failure cases indicate that further improvements may require stronger semantic priors, more adaptive stochastic schedules, or architectures with better long-range context modeling. We leave these directions for future work.

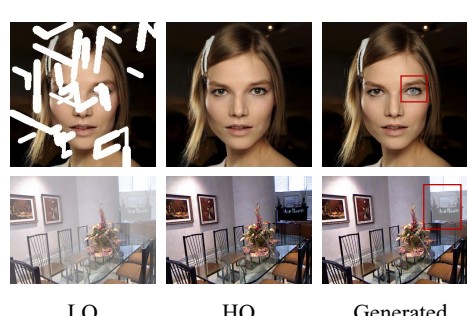

LQ HQ Generated

Figure A1: Representative failure cases of FoD. Red boxes highlight problematic regions.

## D.7 Additional Results for Image Restoration

In this section, we provide additional qualitative results for the four image restoration tasks considered in the main paper, including image deraining, low-light enhancement, image dehazing, and face inpainting. Specifically, Figure A2 compares different sampling strategies, Figure A3 visualizes intermediate FoD sampling states, and Figure A4, Figure A5, Figure A6, and Figure A7 show task-specific restoration examples.

These examples provide a visual complement to the quantitative results in the main paper. Overall, FoD tends to preserve the structural content of the degraded input while recovering sharper local details and more natural textures. The additional samples also illustrate the behavior of the proposed stochastic forward process across different degradation types, including rain streaks, low-light regions, haze, and masked areas.

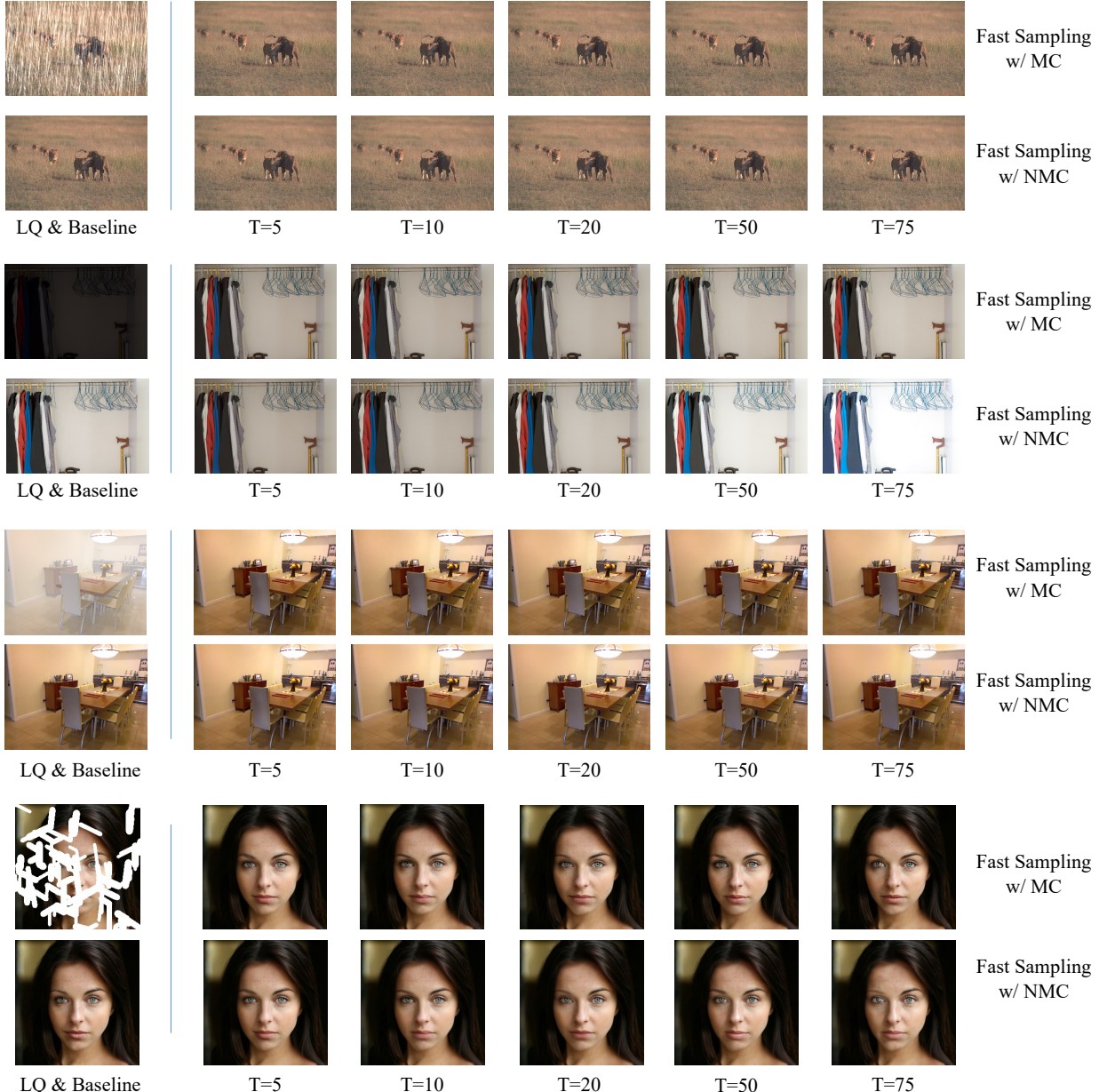

Figure A2: Comparison of fast sampling with Markov and non-Markov chains with different numbers of steps.

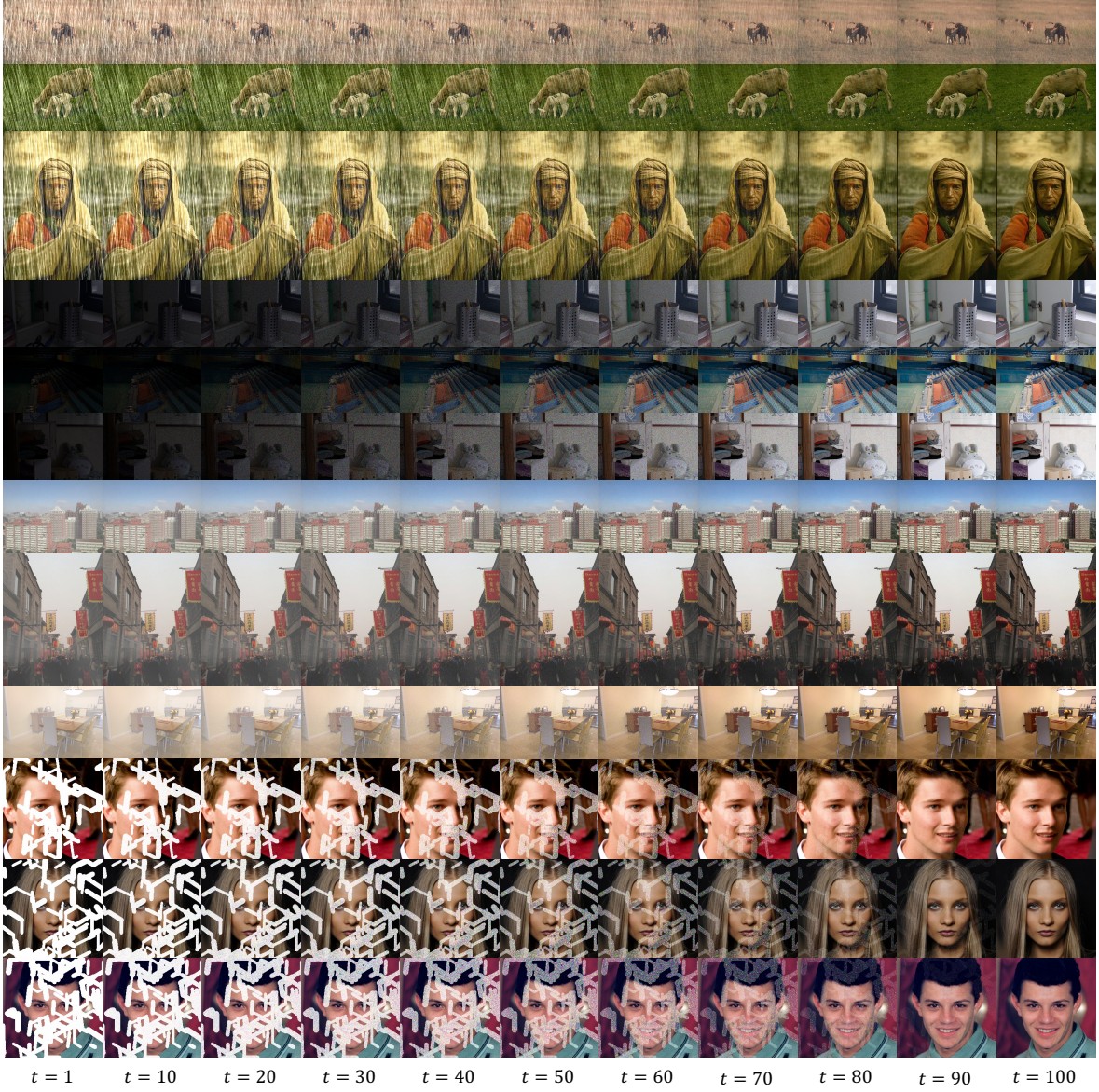

Figure A3: Visualization of the diffusion process using trained FoD models on various tasks, including deraining, dehazing, low-light enhancement, and face inpainting. In each case, FoD gradually injects noise into the degraded regions and subsequently denoises these intermediate states, restoring images with enhanced and corrected details.

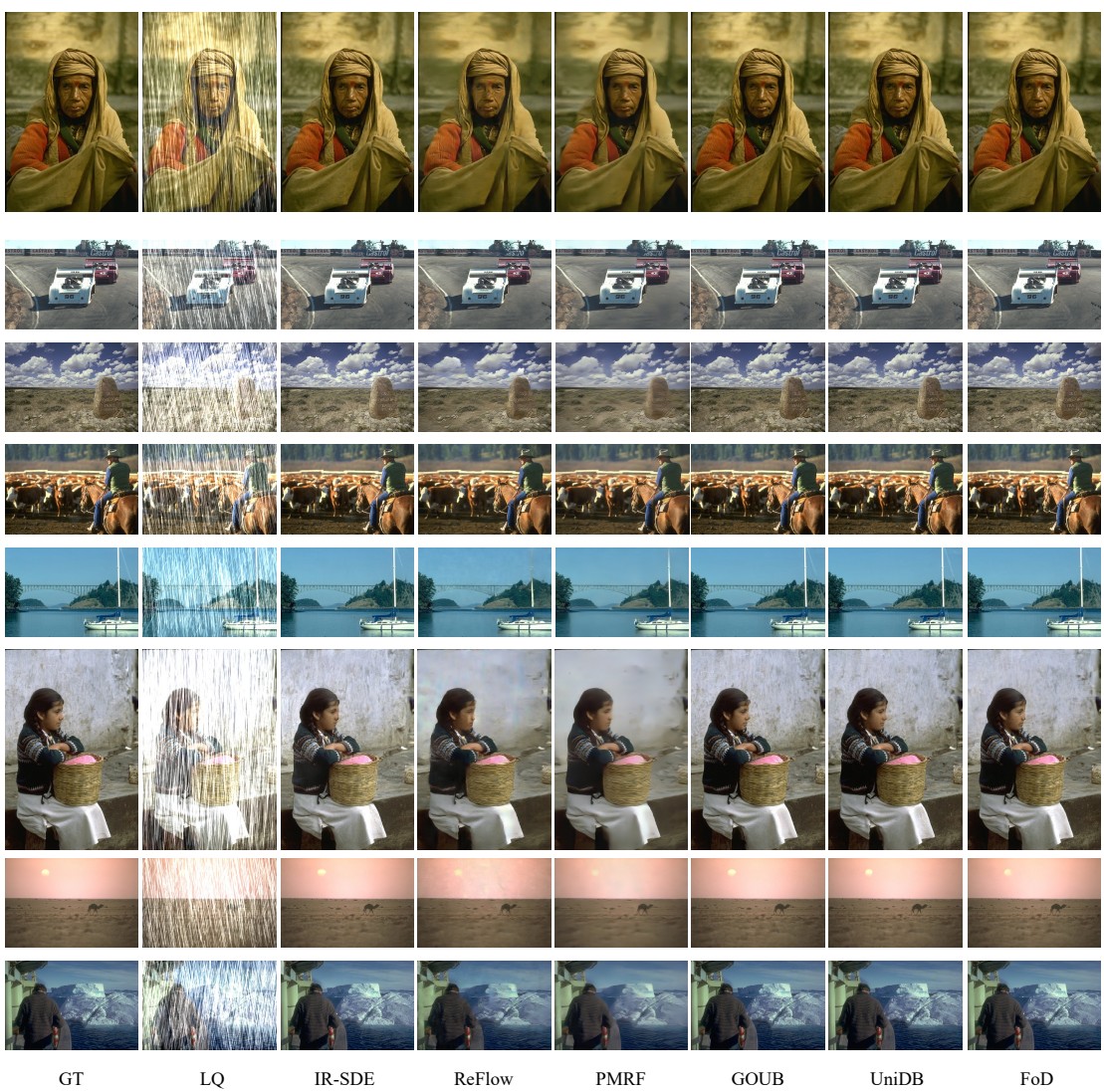

GT  LQ  IR-SDE  ReFlow  PMRF  GOUB  UniDB  FoD

Figure A4: Visual results of image deraining on the Rain100H (Yang et al., 2017) dataset.

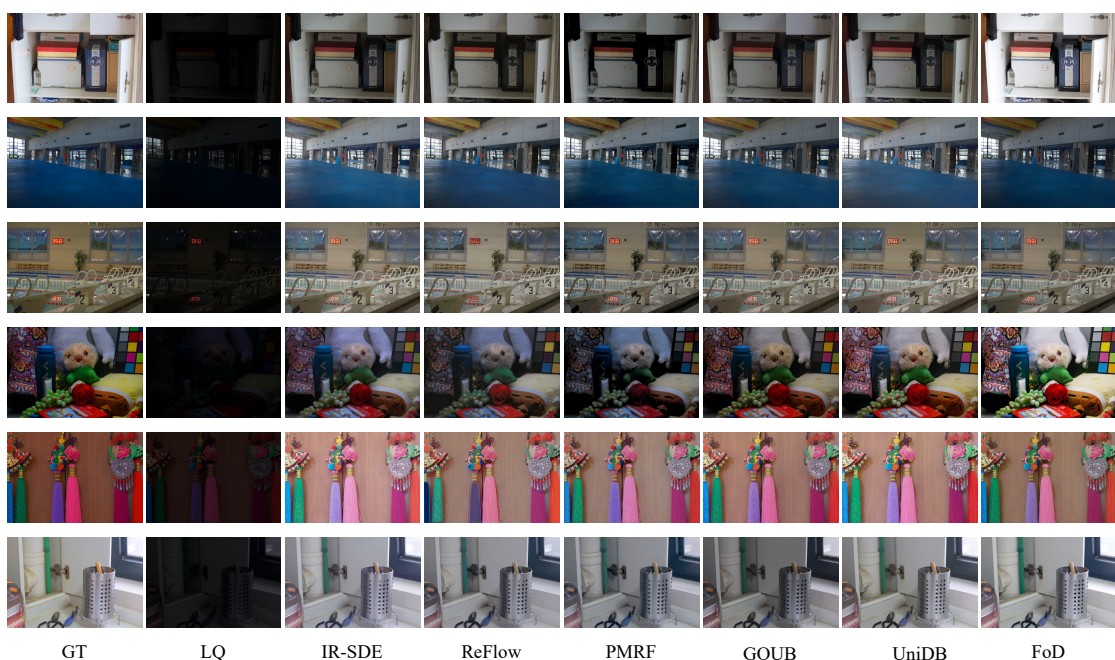

GT      LQ      IR-SDE      ReFlow      PMRF      GOUB      UniDB      FoD

Figure A5: Visual results of image low-light enhancement on the LOL (Wei et al., 2018) dataset.

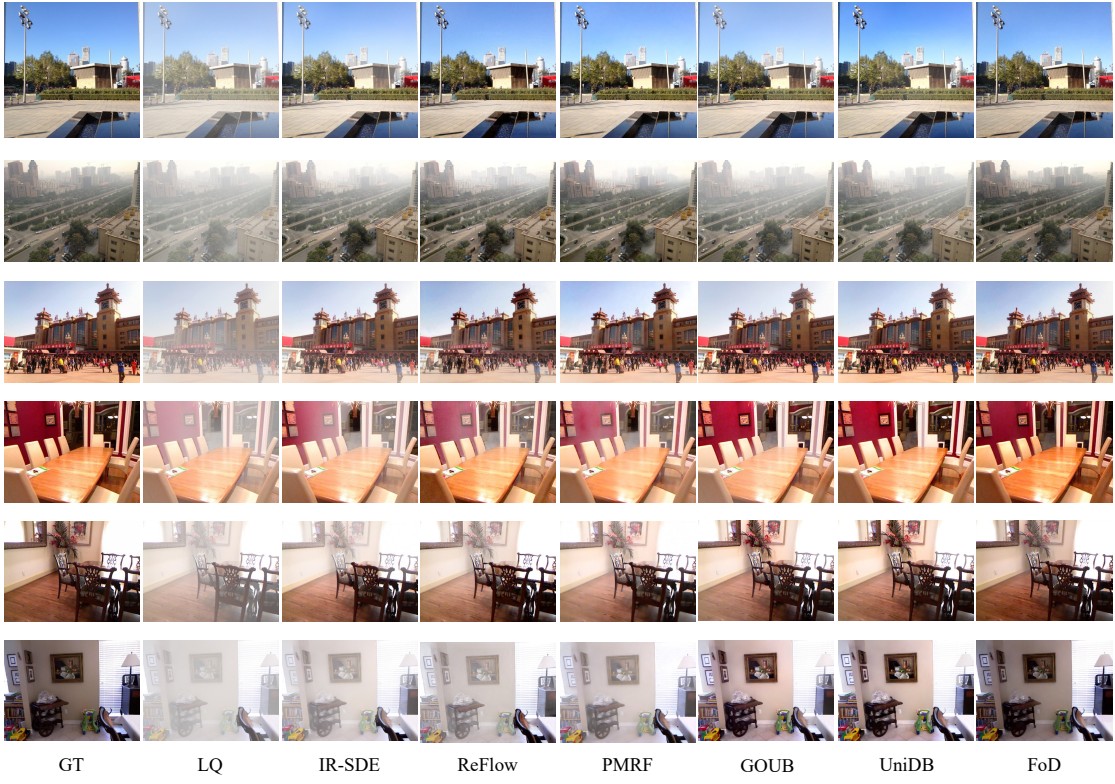

GT      LQ      IR-SDE      ReFlow      PMRF      GOUB      UniDB      FoD

Figure A6: Visual results of image dehazing on the RESIDE-6k (Qin et al., 2020) dataset.

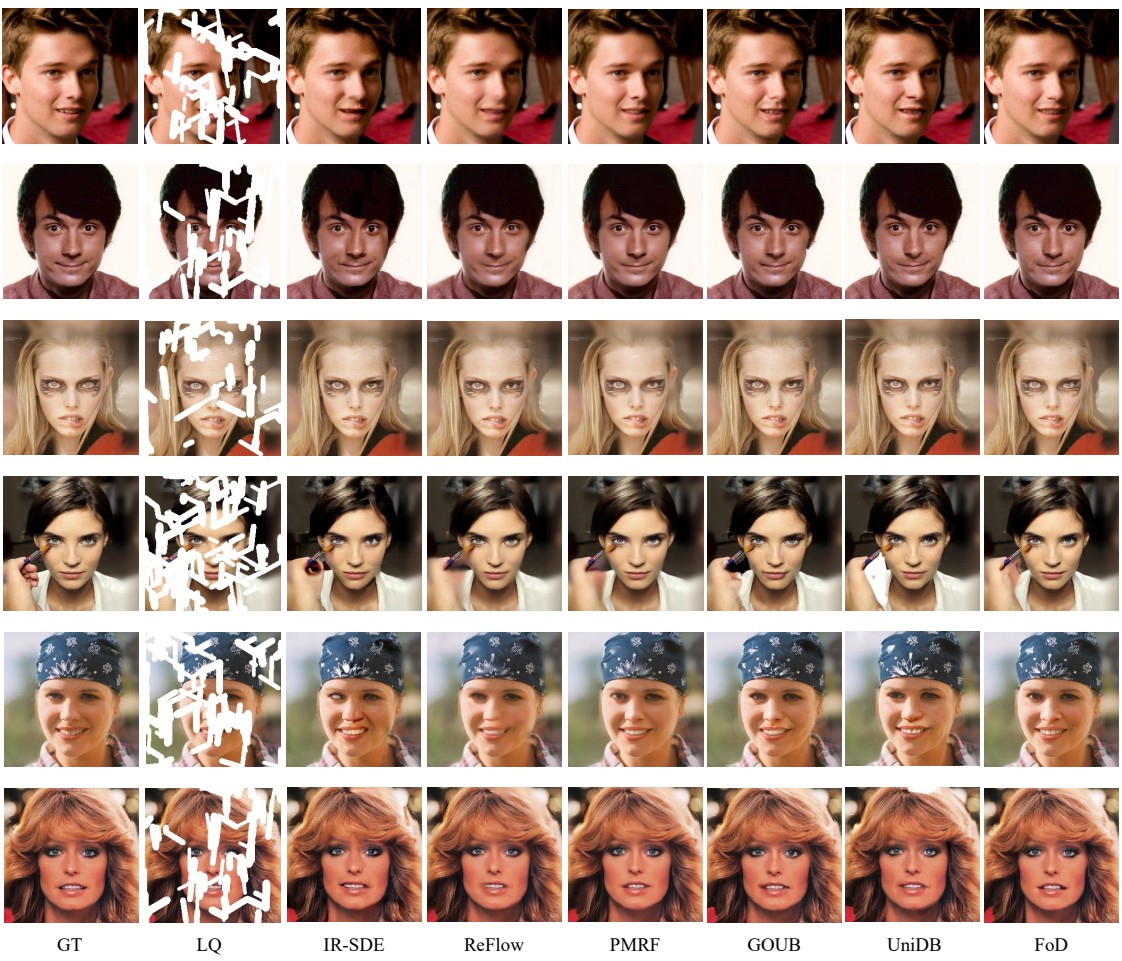

GT LQ IR-SDE ReFlow PMRF GOUB UniDB FoD

Figure A7: Visual results of image inpainting on the CelebA-HQ (Karras et al., 2017) dataset.

