# OpenReview forum: "Efficient Image Restoration with State-Dependent Forward Diffusion"
_TMLR — Under review for TMLR_

### Review · Reviewer_pkVV · 2026-06-11

**Summary Of Contributions:**

This paper proposes an efficient image restoration method with state-dependent forward diffusion. The method designs the mean-reverting stochastic differential equation, and emphasizes the mean-reverting structure in the diffusion function. It can simulates the data transition between source and target distributions, without the need to learn a reverse-time score model. The forward diffusion process is analytically tractable and follows a multiplicative stochastic structure. The experiments are conducted on multiple datasets and demonstrate the effectiveness of the proposed method.

Strength:
1. The presentation of this paper is clear.
2. The proposed method is theoretically sound.
3. Experimental results demonstrate the effectiveness of the proposed approach.

Weakness:
1. The derivation of the method require some assumptions. Some assumptions (e.g., the Gaussian noise assumption) may be standard but not realistic for real-world images, thus the method may be less effective as the theory claims.
2. It is unknown if the experimental results are statistically significant.

**Audience:**

Yes

**Audience Explanation:**

The paper would be of interest to the research community on image restoration, although more general impact may be limited.

**Broader Impact Concerns:**

There are no major concerns that require adding a broader impact statement.

**Claims And Evidence:**

Yes

**Claims Explanation:**

The derivation of the method is back up by theories. It appears to be correct. Experimental results also demonstrate the effectiveness of the proposed approach. There are experiments demonstrating the sampling efficiency and restoration quality of images on different settings.

**Requested Changes:**

Not sure if this is feasible for the image restoration setting, but the current experimental results are not paired with significant test. Experimental results paired with significance test may be more convincing.

It will be better if the authors can explictly list the assumptions or requirements of each theorem, and briefly discuss whether these conditions can be met (or partially met) in real-world settings. In this way, the applicability and limitation of the proposed method can be better understood. It will also be better if some failed image restoration cases can be provided, and discussion on these edge cases can help better understand its limitation.

---

> ### Author Response · Authors · 2026-07-08
> **Response to Reviewer pkVV**
>
> Thank you for your thoughtful review and positive feedback. We appreciate your constructive comments and address them point by point below.
>
> > Q1. The derivation of the method requires some assumptions.
>
> Thank you for the suggestion. We have clarified the assumptions behind the FoD formulation and theoretical results in Appendix A. In particular, the closed-form transition is derived under the It$\hat{\mathrm{o}}$ interpretation with independent Gaussian perturbations and predefined positive schedules. We also note that these assumptions provide a tractable stochastic restoration path, but may not fully capture complex real-world degradations.
>
>
> > Q2. It is unknown if the experimental results are statistically significant.
>
> Following prior iterative restoration methods such as IR-SDE, GOUB, and UniDB, our main evaluation generates one HQ output for each LQ test image. We have clarified this protocol in the appendix. We agree that reporting confidence intervals over multiple random samples or random seeds would provide a more complete uncertainty analysis, especially for small test sets such as LOL. We therefore avoid claiming statistical significance beyond the reported standard evaluation protocol, and have added this point to the limitations.
>
> We have also expanded the limitations to clarify that the current experiments focus on paired and relatively well-defined degradations, and that performance under more complex real-world degradations remains future work. In addition, we also provide additional failure cases in Appendix D.6 to better illustrate the empirical behavior of FoD.

---

### Review · Reviewer_TjaX · 2026-06-21

**Summary Of Contributions:**

## **Strengths**
1. This paper proposes a novel state-dependent SDE (Stochastic Differential Equation) method FoD (Forward Diffusion) by introducing a mean-reverting term to both the drift and diffusion functions in a single state-dependent diffusion process.
2. This paper has the clear algorithms to make the pipeline of FoD more clear and the theory also supports the proposed stochastic flow matching training objective.
3. The authors introduced Markov and non-Markov chain sampling strategies can improve the generated samples’ quality. This is a practical advantage when the method deploys in the real-world scenario.
4. The experiment of FoD across multiple image restoration tasks outperforms other diffusion method in both distortion and perceptual metrics.
## **Weakness**
1. The authors derive their stochastic flow matching loss from an exact KL divergence objective in log-space, but discard the log terms in practice due to numerical instability by relying on a first-order Taylor expansion. This weakens the theoretical proof.
2. How about the performance of the FoD performance when meet the attention-heavy architectures? The experiment in this paper only show the U-Net architecture.
3. The paper primarily evaluates spatial and well-defined corruptions. It remains unclear how FoD performs when subjected to frequency filters, such as low-pass filtering or JPEG compression, which are extremely common in real-world scenarios.

**Audience:**

Yes

**Audience Explanation:**

I think the motivation of this work is really compelling. The proposed method demonstrates highly decent visual and quantitative results on several standard image restoration benchmarks. Although I believe the empirical evaluation currently lacks certain critical experiments (e.g., robustness against frequency-domain degradations), the core methodology is innovative. The findings will certainly be of interest to TMLR readers working on efficient diffusion models and image restoration.

**Broader Impact Concerns:**

High-performance restoration models, particularly those operating with state-dependent SDEs that explicitly target and denoise pixel-level spatial residuals, could inadvertently act as powerful purification mechanisms. Specifically, there is a risk that this technology could be misused to wash out invisible digital watermarks, adversarial defensive noise, or unlearnable examples (UEs) that are intentionally added to images to protect user privacy and copyright against unauthorized data scraping or facial recognition.

**Claims And Evidence:**

No

**Claims Explanation:**

The abstract positions this as a simple yet efficient generative framework. However, because the formulation fundamentally requires the degraded LQ image to dictate the starting structure and bounds the perturbation to the residual between LQ and HQ images, it is strictly an image-to-image restoration tool. It cannot perform unconditional generation from pure noise, meaning the broader generative framework claim should be scoped down.
Moreover, the evaluation is currently restricted to localized spatial corruptions. Because FoD explicitly relies on spatial residuals to guide its state-dependent noise injection, its ability to handle global frequency-domain degradations remains an unverified vulnerability. The broad real-world restoration claims cannot be fully supported without validating the model's structural robustness against these frequency shifts.

**Requested Changes:**

## **Major question**

1. Revise the abstract, introduction, and conclusion to accurately reflect the model's capabilities.
2. To support the broad claims regarding real-world image restoration, the authors must include an experimental evaluation or at least a detailed discussion with qualitative examples of FoD's performance on frequency-based corruptions. I request adding experiments on degradations such as JPEG compression artifacts or severe low-pass filtering.
3. I highly recommend providing a small-scale ablation study using an attention-based backbone. If computationally prohibitive, the authors must explicitly acknowledge this architectural limitation and discuss potential scalability challenges in the main text.
4. The paper derives the stochastic flow matching loss from an exact KL objective but relies on a first-order Taylor expansion in practice, which has a high approximation error during early training. It would significantly improve the theoretical transparency of the paper to briefly acknowledge this numerical instability and the surrogate nature of the loss function in the main methodology section, rather than burying it in the appendix.
5. Adding a brief discussion or a small ablation on whether the 100-step constraint disproportionately penalizes the ODE baselines would strengthen the empirical rigor.
## **Minor Question**
1. I am curious about how this formulation might behave in tasks like image editing [1], where the source is not a heavily degraded image, but a standard clean image requiring semantic modifications.
2. Since FoD explicitly utilizes pixel-level spatial residuals to modulate the stochastic noise injection, I am very interested in its potential interaction with imperceptible structured perturbations, such as adversarial attacks or unlearnable examples [2-4].

[1] Progressive Text-to-Image Diffusion with Soft Latent Direction

[2] Unlearnable examples: Making personal data unexploitable

[3] Versatile Transferable Unlearnable Example Generator

[4] Unlearnable Examples for Diffusion Models: Protect Data from Unauthorized Exploitation

**Please prioritize the critical changes and major questions listed above. If your time is limited during the disscusion period, feel free to focus entirely on those core issues. However, if bandwidth allows, I would greatly appreciate hearing your thoughts on these minor points.**

---

> ### Author Response · Authors · 2026-07-08
> **Response to Reviewer TjaX (part 1/2)**
>
> Thank you for your thoughtful review and constructive feedback. Below, we provide our point-to-point response to address your concerns:
>
> > Q1. Revise the abstract, introduction, and conclusion to accurately reflect the model's capabilities.
>
> We have revised the corresponding sections to more accurately scope the claim. In particular, we rewrote the introduction to start from image restoration methods rather than general generative frameworks, and clarified that FoD is positioned as a conditional image restoration method. We also updated the abstract and conclusion accordingly to avoid implying a general unconditional generative framework.
>
>
> > Q2. Experiments on global frequency-domain degradations such as JPEG compression artifacts.
>
> Thank you for the suggestion. We have added an additional JPEG compression experiment to evaluate FoD beyond spatially localized degradations, as shown in **Table 2**. We train our FoD on DIV2K and evaluate on LIVE1 with JPEG quality factor 10. The results show that FoD remains effective under JPEG artifacts, suggesting that our method can also handle this type of frequency-related corruption.
>
> **Table 2.** Results on JPEG compression artifact removal.
>
> | Method | PSNR ↑ | SSIM ↑ | LPIPS ↓ | FID ↓ |
> |--------|--------:|--------:|---------:|-------:|
> | U-Net | 28.25 | 0.7926 | 0.300 | 76.77 |
> | IR-SDE | 28.35 | 0.7947 | 0.299 | 76.75 |
> | GOUB | 28.45 | 0.7954 | 0.295 | 76.47 |
> | UniDB | 28.47 | 0.7958 | 0.297 | 76.46 |
> | ReFlow | 28.46 | 0.7991 | 0.270 | 71.27 |
> | PMRF | 28.47 | 0.8010 | 0.282 | 71.29 |
> | FoD | 28.60 | 0.8002 | 0.271 | **70.98** |
> | FoD w/ MC | **28.64** | 0.8014 | **0.268** | 71.14 |
> | FoD w/ NMC | 28.61 | **0.8026** | 0.269 | 71.16 |
>
>
> > Q3. Experiments on attention-based backbones.
>
> We have added a small-scale ablation with attention-based backbones on face inpainting, as shown in **Table 3**. Specifically, we compare the default UNet with an attention-based variant, where attention is added to low-resolution features, as well as a ViT-based backbone. The results show that adding attention layers to the UNet slightly improves PSNR, SSIM, and LPIPS, suggesting that FoD is compatible with attention modules. The ViT-based backbone performs worse in this setting, indicating that directly scaling FoD to transformer architectures may require more careful design and tuning. We have added a limitation and conducted this discussion to the appendix to clarify the architectural scope and potential scalability challenges.
>
> **Table 3.** Small-scale ablation of FoD with different backbones on face inpainting.
>
> | Method | PSNR ↑ | SSIM ↑ | LPIPS ↓ | FID ↓ |
> |--------|--------:|--------:|---------:|-------:|
> | FoD-UNet | 30.28 | 0.923 | 0.029 | **16.12** |
> | FoD-UNet w/ attn | **30.44** | **0.925** | **0.027** | 16.40 |
> | FoD-ViT | 29.69 | 0.912 | 0.041 | 19.71 |
>
>
> > Q4. Derive their stochastic flow matching loss from an exact KL divergence objective in log-space, but discard the log terms in practice.
>
> Thank you for pointing this out. We have revised the main methodology section (Section 3.3) to avoid suggesting that the stochastic flow matching loss is an exact variational objective. Instead, we now describe it as a first-order surrogate motivated by the log-residual likelihood induced by the FoD transition. We also explicitly mention that the first-order approximation can be loose during early training.
>
>
> > Q5. Adding a brief discussion or a small ablation on whether the 100-step constraint disproportionately penalizes the ODE baselines would strengthen the empirical rigor.
>
> As suggested, we have now added a small ablation that evaluates ReFlow on image deraining with different sampling steps from 100 to 10, as shown in **Table 4**. The results show that ReFlow is nearly unchanged across different step numbers, with a slight degradation when using fewer steps. This means that the 100-step setting does not unfairly penalize the ODE baseline in terms of restoration quality. In contrast, FoD fast sampling benefits from the closed-form stochastic transition and achieves strong restoration quality with substantially fewer steps, as shown in Table 3 and Figure 3 in the paper.
>
> **Table 4.** Comparison of the ODE baseline ReFlow and FoD on image deraining using different sampling steps.
>
> | Method | PSNR ↑ | SSIM ↑ | LPIPS ↓ | FID ↓ |
> |--------|--------:|--------:|---------:|-------:|
> | ReFlow (*T*=100) | 28.36 | 0.871 | 0.152 | 64.81 |
> | ReFlow (*T*=50) | 28.34 | 0.871 | 0.153 | 64.88 |
> | ReFlow (*T*=20) | 28.32 | 0.870 | 0.153 | 64.99 |
> | ReFlow (*T*=10) | 28.31 | 0.869 | 0.153 | 65.06 |
> | FoD (*T*=100) | 32.56 | 0.925 | 0.038 | **14.10** |
> | FoD (*T*=10) | **33.27** | **0.934** | 0.039 | 15.14 |

---

> > ### Author Response · Authors · 2026-07-08
> > **Response to Reviewer TjaX (part 2/2)**
> >
> > > Q6. I am curious about how this formulation might behave in tasks like image editing.
> >
> > We have not evaluated FoD on image editing, as our current focus is image restoration. In addition, image editing is more challenging since it typically requires additional text or semantic conditions to specify the desired modification while preserving irrelevant regions. One possible direction is to incorporate such guidance into the FoD transition or the predicted endpoint, but this requires additional design beyond the current restoration setting.
> >
> > > Q7. Potential interaction with imperceptible structured perturbations.
> >
> > We have not studied this interaction in the current paper. Since FoD uses residual-dependent stochastic perturbations, imperceptible structured perturbations may affect the learned residual field or be partially suppressed by the restoration process. Thank you for raising this point and we will consider this an important direction for future robustness analysis.
> >
> > > Q8. Broader impact statement
> >
> > We have added a broader impact statement noting that restoration models may potentially affect imperceptible protective signals, such as watermarks or adversarial perturbations, and leave this robustness analysis to future work.

---

### Review · Reviewer_sNrZ · 2026-06-26

**Summary Of Contributions:**

This paper presents FoD, a state-dependent mean-reverting forward SDE framework for paired image restoration. Unlike conventional diffusion-based approaches that rely on coupled forward-backward processes, FoD performs restoration through a single forward SDE by introducing mean-reversion terms into both the drift and diffusion functions. The key technical contributions are: (1) a closed-form analytical solution to the proposed SDE, revealing a multiplicative stochastic structure; (2) a stochastic flow matching training objective motivated from the FoD transition; and (3) efficient few-step sampling strategies via Markov and non-Markov chain formulations based on the tractable transition. Experiments on four image restoration tasks — deraining, dehazing, low-light enhancement, and face inpainting — show improvements on most reported metrics over IR-SDE, GOUB, UniDB, ReFlow, PMRF, and a U-Net baseline.

Key strengths:

- The mathematical formulation is clean and elegant. The state-dependent mean-reverting SDE is a well-motivated modification of the classical mean-reverting SDE, and the closed-form transition is useful for both analytical understanding and practical sampling.
- The connection between FoD and broader families of stochastic interpolants and flow matching is clearly articulated, helping position the contribution within the existing theoretical landscape.
- The ablation on noise injection effectively demonstrates the importance of stochasticity in image restoration, with large performance gaps between FoD and its noise-free variant across all reported tasks.
- The visualization of the diffusion process provides useful intuition about FoD's behavior: stochastic perturbations appear to be concentrated more strongly in degraded regions, which is a desirable property for restoration.
Key weaknesses:

- The paper emphasizes "efficiency" prominently, including in the title and abstract, but does not provide direct wall-clock inference time, NFE, parameter count, or memory usage comparisons against the baselines.
- The log-residual formulation in Corollary 3.2 uses $\log(\mu - x_t)$, which is not generally well-defined componentwise when $\mu - x_t$ is negative; the appeal to sign consistency does not resolve this mathematical issue.
- The convergence claim — that the mean-reverting structure "guarantees convergence to fixed clean points" — is stated strongly, but the implementation relies on a finite-time approximation with a chosen exponential factor $\delta = 0.001$, and the stochastic component of the terminal residual is not analyzed.
- The empirical claims should be more carefully worded, since FoD is not best on every metric and the few-step samplers involve distortion/perceptual trade-offs.
- The variational motivation relies on a first-order Taylor approximation that is loose during early training, as shown by the authors' own analysis in Table A1.

**Additional Comments:**

Overall, I find the paper promising. The proposed FoD process is conceptually clean, the closed-form transition is useful, and the empirical results are strong across several restoration tasks. My main concerns are about the precision of the theoretical statements and the strength of the efficiency and empirical claims. Addressing the convergence/log-residual issues, adding direct computational comparisons, and improving reproducibility details would substantially strengthen the submission.

**Audience:**

Yes

**Audience Explanation:**

Yes. The paper addresses the intersection of SDE-based generative modeling, flow matching, stochastic interpolants, and image restoration, all of which are relevant to the TMLR audience. The idea of using a forward-only stochastic process that preserves noise injection while avoiding reverse-time score estimation is conceptually appealing and could inspire extensions to other paired conditional generation and restoration tasks. The closed-form transition and few-step sampling procedures also make the method practically interesting.

The work should be of interest to researchers studying generative restoration algorithms, stochastic-process formulations of conditional generation, and efficient alternatives to diffusion-bridge or reverse-time score-based methods. The empirical coverage across multiple restoration tasks further increases the paper's relevance.

**Broader Impact Concerns:**

No significant ethical concerns that would prevent publication are identified. The paper focuses on standard image restoration benchmarks, and the method is not specifically designed for malicious image manipulation. However, because FoD is a generative restoration method and includes face inpainting experiments, a brief broader impact statement would be appropriate. It should acknowledge that generative restoration models may hallucinate plausible but incorrect details in heavily degraded or masked regions, and that restored images should not be treated as ground truth in high-stakes settings such as forensics, surveillance, identity verification, legal evidence, or medical imaging. The potential dual-use risk of face restoration/inpainting methods for misleading image edits could also be briefly noted.

**Claims And Evidence:**

Yes

**Claims Explanation:**

Yes, with reservations.

The main theoretical claims are mostly supported. The proposed SDE is clearly stated, the closed-form solution is derived, and the appendix provides formal proof for the multiplicative stochastic structure. The experimental evidence also supports the central claim that FoD is competitive with, and often better than, the selected diffusion, diffusion-bridge, and flow-matching baselines across several image restoration tasks. The ablations on noise injection, schedule choices, and sampling strategies are useful and generally convincing within the scope of the chosen comparisons.

However, several claims should be qualified or supported more carefully.

First, the efficiency claim is not fully established beyond the internal comparison between FoD's 100-step Euler–Maruyama sampler and its 10-step MC/NMC samplers. Since the title and abstract emphasize efficiency, the paper should provide direct computational comparisons against the baselines, including wall-clock inference time, NFE, memory usage, and parameter counts under the same hardware and resolution.

Second, the convergence claim needs more precise treatment. The paper states that the mean-reverting structure "guarantees the convergence to fixed clean points," but the actual implementation uses a finite-time horizon where the deterministic exponential factor is set to $\delta = 0.001$, not zero. Moreover, the stochastic exponential factor — the $\bar{\sigma}_T \epsilon$ term in Proposition 3.1 — means the terminal state $x_T = (x_0 - \mu) \exp(\bar{m}_T + \bar{\sigma}_T \epsilon) + \mu$ retains random variation even when $\exp(\bar{m}_T)$ is small. A bound on the expected terminal residual and its variance would make this claim more precise.

Third, the log-residual formulation in Corollary 3.2 contains a mathematical imprecision. The corollary writes $\log(\mu - x_t)$, but $\mu - x_t$ can be negative componentwise. The appendix derivation uses $\ln|x_t - \mu|$ before dropping the absolute value by appealing to sign consistency. Sign preservation through the GBM structure is valid, but it does not make the logarithm of a negative quantity well-defined. The practical loss in Eq. 11 is unaffected since it does not involve logarithms, but the theoretical statement should be mathematically precise.

Fourth, the empirical claim of superior performance should be stated more carefully. FoD is best on most reported metrics, but not all: for example, PMRF obtains higher PSNR than FoD on face inpainting. Similarly, the few-step MC/NMC samplers improve many distortion metrics but sometimes degrade perceptual metrics such as LPIPS and FID relative to the 100-step EM sampler. These are reasonable trade-offs, but they should be stated explicitly rather than implied as uniform improvement.

Finally, the variational motivation for the stochastic flow matching loss should be presented more cautiously. The first-order Taylor approximation is explicitly loose during early training, so the loss is better described as a stable surrogate motivated by the FoD transition rather than as an exact variational objective.

**Requested Changes:**

**Critical to securing my recommendation for acceptance:**

1. **Substantiate or temper the efficiency claim.** Since efficiency is central to the paper, including the title and abstract, please provide a direct computational comparison between FoD and the main baselines under the same hardware, batch size, and image resolution. At minimum, please report wall-clock inference time, number of function evaluations (NFE), model parameter count, and GPU memory usage. The current 10-step MC/NMC results are useful, but they are mainly internal comparisons against FoD's own 100-step Euler–Maruyama sampler and do not establish whether FoD is more efficient than IR-SDE, GOUB, UniDB, ReFlow, or PMRF at comparable quality levels.
2. **Fix the signed log-residual formulation.** Corollary 3.2 and the variational explanation use expressions such as $\log(\mu - x_t)$, but $\mu - x_t$ can be negative componentwise. The appendix first derives the result with $\ln|x_t - \mu|$, but then drops the absolute value by appealing to sign consistency. Sign preservation through the GBM structure is valid, but it does not make the logarithm of a negative value well-defined. Please rewrite the result using $\log|\mu - x_t|$ together with a separate sign-preservation statement, or define a signed-log representation explicitly. The practical loss in Eq. 11 is unaffected, but the theoretical statement should be mathematically precise.
3. **Clarify the convergence claim.** The paper states that the mean-reverting structure "guarantees convergence to fixed clean points." Please specify the conditions under which this holds. In the finite-time implementation, the deterministic exponential factor is set to a small nonzero value $\delta = 0.001$, and the stochastic exponential factor $\bar{\sigma}_T \epsilon$ remains present. A short explanation or bound for the expected terminal residual $\mathbb{E}[x_T - \mu]$ and its variance under the chosen schedules would make the claim much clearer. Alternatively, please distinguish asymptotic convergence from finite-time approximate convergence.
4. **Temper claims of uniform superiority and no quality compromise.** FoD is best on most reported metrics, but not all; for example, PMRF achieves higher PSNR than FoD on face inpainting. Similarly, the few-step MC/NMC samplers improve many distortion metrics but sometimes degrade perceptual metrics such as LPIPS and FID relative to the 100-step EM sampler. These are reasonable trade-offs, but the paper should state them explicitly rather than implying uniform improvement across all metrics and samplers.
5. **Improve essential reproducibility details.** Please specify the image resolution or crop size used for each dataset during training and testing, the batch size, model parameter count, and whether stochastic evaluation metrics are computed from one random sample per input, an average over multiple samples, or another protocol. These details directly affect the interpretation of the reported metrics.
**Changes that would strengthen the work:**

6. **Clarify the practical scope relative to non-generative restoration methods.** The current comparisons are mainly to diffusion, diffusion-bridge, and flow/rectified-flow approaches, plus a U-Net baseline. This is a reasonable scope, but the paper should either add one or two representative strong regression-based restoration baselines, such as Restormer or NAFNet, or explicitly state that the empirical comparison is limited to generative/diffusion/bridge/flow-matching methods and discuss the missing comparison as a limitation. This would help readers understand FoD's practical trade-off between perceptual quality and distortion metrics.
7. **Discuss uncertainty and stochastic variability.** Reporting standard deviations or confidence intervals, especially for stochastic sampling and small test sets such as LOL, would make the empirical evidence more robust. This is particularly relevant for FID estimation on small sets.
8. **Present the variational motivation more cautiously.** The first-order Taylor approximation used in the variational explanation is loose during early training, as the appendix already acknowledges. The main text should avoid suggesting that the stochastic flow matching loss is an exact variational objective; it would be more accurate to describe it as a stable surrogate motivated by the FoD transition.

---

> ### Author Response · Authors · 2026-07-08
> **Response to Reviewer sNrZ (part 1/2)**
>
> Thank you for the insightful review and constructive comments, which accurately summarize our paper and our intended contributions. We appreciate your feedback and are happy to provide our point-to-point response below:
>
> > Q1. Substantiate or temper the efficiency claim.
>
> Thank you for pointing this out. We have now added an efficiency comparison (on the image deraining test set) in terms of wall-clock inference time, NFE, parameter count, and PSNR, as shown in **Table 1**. The results show that our few-step sampler, _FoD w/ MC_, achieves the best PSNR performance while using only 10 NFEs per image, giving about 9x speedup compared with other flow matching based approaches under the same framework and memory usage. It is also noted that IR-SDE, GOUB, and UniDB are implemented under the same framework and backbone, while ReFlow, PMRF, and our FoD are implemented under another shared framework (guided-diffusion). We have added more details in the revised manuscript and clearly state that FoD mainly improves sampling efficiency through its closed-form transition.
>
> **Table 1.** Efficiency comparison in terms of wall-clock inference time, NFE, and parameter count on deraining.
>
> | Method | Wall-clock time | NFE | Parameter count | PSNR ↑ |
> |--------|----------------:|----:|----------------:|--------:|
> | IR-SDE | 11.59s | 100 | 137.15M | 31.65 |
> | GOUB | 20.11s | 100 | 137.15M | 31.96 |
> | UniDB | 20.17s | 100 | 137.15M | 32.05 |
> | ReFlow | 3.86s | 100 | 73.45M | 28.36 |
> | PMRF | 3.89s | 100 | 36.22M + 73.45M | 29.01 |
> | FoD (Ours) | 3.85s | 100 | 73.45M | 32.56 |
> | **FoD w/ MC (Ours)** | **0.43s** | **10** | **73.45M** | **33.27** |
>
> > Q2. Fix the signed log-residual formulation.
>
> We have now revised Corollary 3.2 to explicitly write $\log |\mu - x_t|$ and clarified that the sign information is preserved. For a given sample, the element-wise sign of $\mu-x_t$ remains consistent across all times $t$, as the FoD transition only multiplies $\mu-x_s$ by a strictly positive stochastic exponential factor (as shown in the closed-form FoD solution).
>
> > Q3. Clarify the convergence claim.
>
> Thanks for the suggestion. We have revised the convergence claim to "the mean-reverting structure drives the low-quality data toward the clean endpoint with controlled stochastic variation", rather than guaranteeing exact finite-time convergence. From the FoD solution, the terminal residual satisfies $x_T - \mu = (x_0-\mu) \exp\left(-\int_0^T(\theta_t+\frac{1}{2}\sigma_t^2) \mathrm{d}t + \int_0^T \sigma_t  \mathrm{d}w_t \right)$. Since the term $\int_0^T \sigma_t \mathrm{d}w_t \sim \mathcal{N}\left(0,\int_0^T\sigma_t^2 \mathrm{d}t\right)$, we obtain the expected terminal residual $\mathbb{E}[x_T - \mu] = (x_0 - \mu) \exp\left(-\int_0^T\theta_t \mathrm{d} t\right)$. Moreover, in the log-residual space used in Corollary 3.2, the log-residual variance is $\int_0^T\sigma_t^2dt$. We also clarify that this corresponds to finite-time approximate convergence with nonzero stochastic variation, rather than exact convergence. We have added these details to Appendix C.3 in the revised manuscript.
>
> > Q4. Temper claims of uniform superiority and no quality compromise.
>
> Thank you for pointing this out. We have revised the manuscript to describe FoD as achieving strong overall performance and favorable efficiency-quality trade-offs, rather than uniformly outperforming all baselines across all metrics. In addition, we also explicitly discuss the observed exceptions and trade-offs. For example, PMRF obtains a higher PSNR on face inpainting, while FoD remains competitive on the other metrics. Similarly, the few-step MC/NMC samplers substantially reduce NFE and improve distortion metrics such as PSNR and SSIM, but can slightly degrade perceptual metrics (LPIPS and FID) compared with the 100-step EM sampler in some cases. We have added these clarifications to make the empirical claims more precise.
>
> > Q5. Improve essential reproducibility details.
>
> We have added the missing reproducibility details in the appendix. Specifically, we adopt the same setting for all restoration tasks: the batch size is 16, all training images are randomly cropped to 256x256, and test images are evaluated at their original resolution. For stochastic evaluation, we follow previous works to generate one HQ output from each LQ test image. We have also reported the implementations used for FID and LPIPS: PyTorch-FID and the image quality assessment (IQA) toolbox, respectively.

---

> > ### Author Response · Authors · 2026-07-08
> > **Response to Reviewer sNrZ (part 2/2)**
> >
> > > Q6. Clarify the practical scope relative to non-generative restoration methods.
> >
> > Thank you for the suggestion. We have revised the manuscript to explicitly state that our empirical comparisons focus on iterative generative restoration methods, including diffusion, diffusion-bridge, and flow/rectified-flow approaches, since our goal is to study stochastic iterative restoration and few-step sampling with a state-dependent forward SDE. We have also added a limitation noting that strong regression-based restoration networks are not included in the current comparison.
> >
> > > Q7. Discuss uncertainty and stochastic variability.
> >
> > In this work, we follow the evaluation protocol used by prior restoration methods such as IR-SDE, GOUB, and UniDB: for each LQ test image, we generate only one corresponding HQ output and compute all metrics on this generated test set. Under this one-sample-per-input protocol, the main tables are directly comparable to prior work, but they do not estimate variability across multiple stochastic samples or random seeds. We have clarified this in the appendix and added a limitation noting that FID on small test sets can have high estimation variance.
> >
> > > Q8. Present the variational motivation more cautiously.
> >
> > Thank you for the suggestion. We have revised the main text to avoid suggesting that the stochastic flow matching loss is an exact variational objective. Instead, we now describe it as a stable first-order surrogate motivated by the log-residual likelihood induced by the FoD transition.
> >
> > > Q9. Broader impact statement
> >
> > We have added a brief broader impact statement discussing possible hallucinated details in generative restoration, especially for face inpainting, and noting that restored images should not be used as reliable evidence in high-stakes settings such as such as forensics, surveillance, identity verification, legal evidence, or medical imaging.